# AutoQG: An Automated Framework for Evidence-Traceable Question Generation via Ontology-Guided Knowledge Graph Construction

## Abstract

Large Language Models (LLMs) present unprecedented opportunities for generating scientific questions. However, existing approaches face two key limitations: heavy reliance on costly human annotations and the production of brittle, unverifiable outputs. To address these challenges, we propose **AutoQG**, a fully automated multi-agent framework for evidence-grounded scientific QA generation. AutoQG comprises three complementary agents: (i) **KG Extraction Agent**, which performs ontology-guided knowledge graph construction with section-aware prompts for precise information retrieval; (ii) **KG Evaluation Agent**, a multi-dimensional evaluation module with iterative refinement to ensure accuracy and consistency; and (iii) **QA Generation Agent**, which produces schema-constrained QA pairs grounded in reasoning paths and explicit textual evidence. Applied to over 4,000 scientific papers, AutoQG constructs 243k triples and introduces $AutoQG^{20k}$, a benchmark containing more than 20,000 QA pairs. Each pair is explicitly linked to its reasoning chains and supporting evidence, ensuring transparency and verifiability. We further release $AutoQG^{7k}$, a challenging subset designed with hard questions that strong LLMs struggle to answer. Extensive experiments demonstrate that AutoQG consistently outperforms strong baselines in both human evaluation and LLM-as-a-Judge assessments. By transforming LLM output into a controlled and auditable pipeline, **AutoQG** advances evidence-based AI for the understanding of reliable scientific knowledge. Source code will be released upon publication.

## 1 Introduction

Large Language Models (LLMs) have demonstrated remarkable capabilities in text comprehension and knowledge extraction, presenting a transformative opportunity for navigating the expansive and complex domain of scientific literature (OpenAI, 2023; Grattafiori et al., 2024; Reid et al., 2024). Yet, this promise is met with a critical challenge when applying these models to scientific texts: knowledge in this domain is inherently distributed. A key claim introduced in the abstract is often detailed in the methods section and validated by evidence in the results. Existing approaches frequently fail to synthesize this cross-sectional information into coherent reasoning chains, leading to question-answering (QA) pairs that lack traceability and robust evidentiary support (Lee et al., 2025; Wang et al., 2025). Consequently, without a principled framework to constrain their behavior, LLM outputs often degrade into brittle factual recall or unverifiable claims, falling short of the methodological rigor and auditability demanded by scientific inquiry (Schryen et al., 2025). This gap highlights the urgent need for frameworks that transform the generative power of LLMs into a controlled, verifiable, and evidence-grounded process, with a particular focus on generating high-fidelity scientific QA pairs that capture reasoning chains and provide traceable evidence for knowledge claims.

Recent progress in scientific QA Generation has been defined by a trade-off between quality and scalability. On one hand, manual annotation by domain experts yields high-quality benchmarks but is costly and difficult to scale (Lála et al., 2023; Asai et al., 2024). On the other, automated methods, while scalable, often produce shallow QA pairs that lack verifiability and prioritize factual recall over higher-order reasoning (Auer et al., 2023; Wang et al., 2024). Even recent LLM-based frameworks

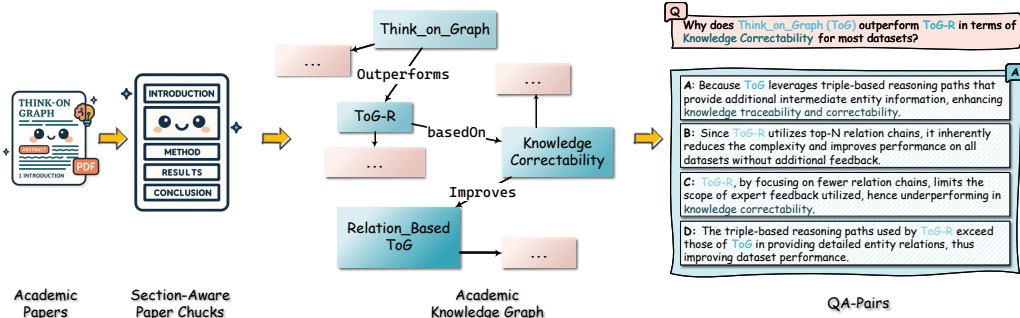

Figure 1: An illustrative overview of our pipeline. The process begins with an academic paper, which is segmented into section-aware chunks and transformed into a structured knowledge graph capturing the paper's core entities and relations. From this graph, we derive evidence-traceable QA pairs by linking each question and answer directly to its underlying reasoning path and supporting text.

have yet to fully resolve this tension, as they are often not specifically tailored for the structured and evidence-centric nature of scientific texts (Vladika & Matthes, 2024; Fang et al., 2024).

To bridge this gap between scalability and rigor, we posit that an intermediate structured representation is necessary, one that can explicitly model the reasoning paths and evidentiary links within a document. A knowledge graph (KG) is an ideal scaffold for this purpose (Lan et al., 2021). By transforming unstructured text into a network of entities and their relationships (Zhang et al., 2022), a KG provides an auditable backbone for generating complex questions that require multi-hop reasoning (Dong et al., 2023) and are explicitly traceable to the source text (Yih et al., 2015). This principle is illustrated in Figure 1, which shows how a structured knowledge graph serves as the foundation for generating evidence-traceable QA pairs from scientific papers.

To overcome these limitations, we present **AutoQG**, a principled, fully-automatic multi-agent framework that transforms pretrained LLMs into a controllable, evidence-traceable pipeline. First, the **KG Extraction Agent** constructs a knowledge graph using section-aware prompts to navigate the intricate structure of scientific literature. This process is ontology-guided: it uses an automatically induced ontology as a guide to constrain the LLM's output, rather than relying on a rigid, predefined ontology. However, this flexible approach requires a rigorous validation step, which motivates our second component: **KG Evaluation Agent**. This agent serves as a critical quality control mechanism, performing a multi-dimensional evaluation and providing diagnostic feedback in an iterative refinement loop that progressively improves the graph's accuracy and consistency. Finally, once the graph is validated, the **QA Generation Agent** transforms the refined knowledge into verifiable QA pairs. It identifies multi-hop reasoning paths and applies schema-constrained prompting, ensuring the generated questions are explicitly traceable to their reasoning chains and source evidence, which in turn allows for fine-grained control over reasoning complexity.

In summary, our key contributions are threefold:

- We propose **AutoQG**, an automated multi-agent framework that enables high-fidelity scientific QA generation by integrating ontology-guided and section-aware KG extraction, multi-dimensional evaluation, and schema-constrained QA generation.

- We introduce two large-scale, evidence-traceable QA benchmarks derived from over 4,000 scientific papers: $AutoQG^{20k}$, containing more than 20k QA pairs explicitly linked to reasoning paths and supporting evidence, and the more challenging $AutoQG^{7k}$, which focuses on hard scientific questions that current LLMs struggle to answer.

- Through extensive experiments combining human and LLM-as-a-Judge evaluations, we demonstrate that **AutoQG** consistently outperforms strong baselines. Furthermore, extensive ablation studies quantify the significant contribution of each component of our framework.

## 2 RELATED WORK

**Scientific Comprehension Benchmarks.** In evaluating the automatic comprehension of academic papers, benchmark datasets are increasingly highlighting both domain knowledge and analytical understanding. Widely used datasets such as emrQA (Pampari et al., 2018; Jin et al., 2019; Saikh et al., 2022; Shamsabadi et al., 2024; Wang et al., 2024; Wellawatte et al., 2025; Phan et al., 2025) often treat paper content as independent text segments, overlooking broader structural relationships within academic articles and thus leaving certain information unutilized. Other datasets (Lála et al., 2023; Bai et al., 2025; Skarlinski et al., 2024) adopt direct text-matching strategies without deeper semantic modeling. While this design is well-suited for evaluating retrieval or surface-level comprehension, it may be less effective for assessing tasks that demand higher-order reasoning or complex analytical skills.

**Academic QA Generation Methods.** Current academic QA datasets are typically constructed through two main paradigms: crowdsourced human annotation and LLM-based automatic generation. In the crowdsourced approach, all labeling tasks are carried out manually, with quality controlled by expert reviewers (Lála et al., 2023; Bai et al., 2025; Skarlinski et al., 2024; Asai et al., 2024). This method yields high-quality data but requires strong domain expertise, incurs high costs, and makes iterative refinement difficult. By contrast, LLM-based generation leverages prompting frameworks and automated tools (Lee et al., 2023; Wan et al., 2024; Yu et al., 2025) offering scalability but producing datasets that remain surface-level, focusing on factual recall rather than higher-order reasoning or causal analysis.

**Automatic Knowledge Graph Construction.** Recent advances in knowledge graph (KG) construction have largely relied on large language models (LLMs) and pre-trained language models (PLMs), with methods primarily formulated as structured information extraction under few-shot or zero-shot settings (Agrawal et al., 2022; Li et al., 2023a; Han et al., 2023; Li et al., 2024b; Liu et al., 2024; Li et al., 2023b). Structured parsing strategies (Li et al., 2023a; Xue et al., 2024; Sainz et al., 2024; Li et al., 2024a) enable triple generation that implicitly embeds reasoning paths, yet outputs are often constrained by handcrafted templates (Kim et al., 2023; Ding et al., 2024), limited generalization to unseen relations (Zhang et al., 2023), and unstable zero-shot performance in specialized domains (Gutierrez et al., 2022). Benchmark evaluations on BLURB, TACRED, and Re-DocRED (Wadhwa et al., 2023; Xue et al., 2024; Bai et al., 2025; Kim et al., 2025) further reveal that automatic metrics underestimate true performance, as many false positives are semantically valid. While recent studies explore quality-controlled or iterative pipelines (Gao et al., 2025; Wan et al., 2023; Jiao et al., 2023), most approaches still lack explicit reasoning or evidence modeling, limiting their reliability for explainable KG-QA.

## 3 METHOD

### 3.1 OVERVIEW

We propose a multi-agent framework that constructs structured academic knowledge graphs from papers and generates evidence-traceable, difficulty-controllable QA pairs—without manual annotation or model finetuning. As illustrated in Figure 2, the system follows a three-stage *construct–evaluate–generate* paradigm. Each stage is formally defined below.

### 3.2 KG EXTRACTION AGENT: SECTION-AWARE AND ONTOLOGY-GUIDED EXTRACTION

The KG Extraction Agent serves as the entry point of our multi-agent pipeline, transforming raw scientific text into a structured knowledge graph. It leverages the zero-shot generation capabilities of large language models by mapping their abstract understanding of text into concrete, ontology-aligned triples. Scholarly documents are first normalized into a canonical discourse structure. Each section $S_k$ is paired with a prompt template $\Phi_k$ that encodes inductive biases reflecting its discourse role, further constrained by a predefined knowledge ontology. The full prompts are provided in Appendix A.4.

This design stands in clear contrast to flat prompting baselines, which treat the document as a single undifferentiated text stream and often produce sparse or noisy triples. By constraining the extraction process with both discourse-aware prompts and explicit ontology guidance, the agent suppresses

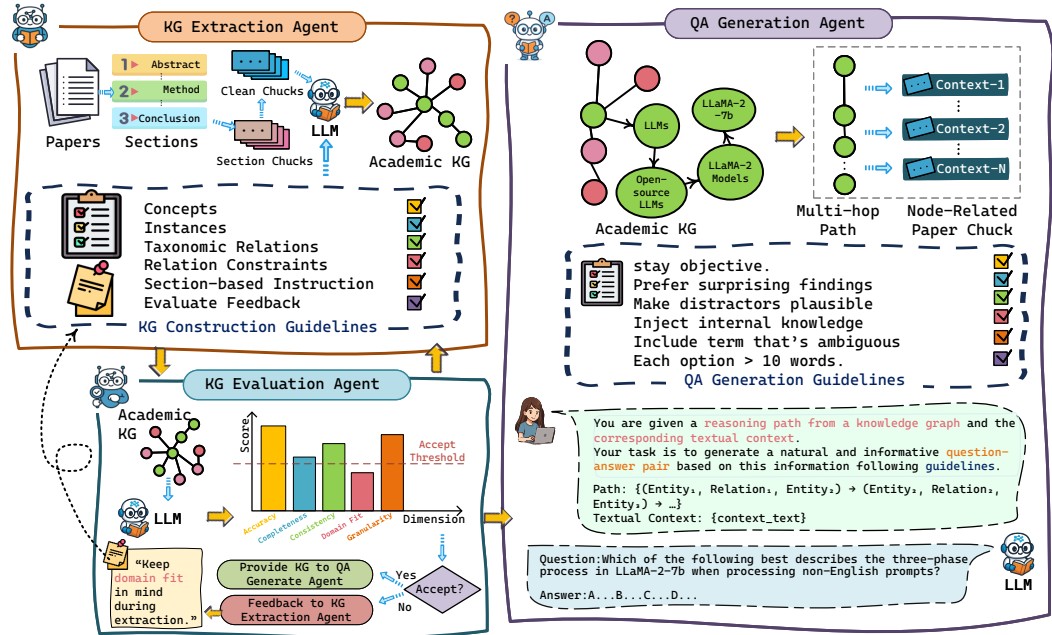

Figure 2: The proposed multi-agent pipeline for constructing academic knowledge graphs and generating multi-hop QA pairs from scientific papers.

spurious entities and improves both density and precision of extracted relations. For instance, consider a sentence from the *Methods* section:

*"We trained Model-A on the CIFAR-100 dataset with a learning rate of 0.001."*

A flat baseline may only extract:

$$(Model\text{-}A,\ trainedOn,\ CIFAR\text{-}100)$$

whereas our section-specific template, guided by predefined relation types in the ontology, yields richer and more precise triples:

$$(Model\text{-}A,\ evaluatedOn,\ CIFAR\text{-}100), \quad (Model\text{-}A,\ hasHyperparameter,\ learning\ rate{=}0.001).$$

To ensure schema alignment and reduce hallucinations, in-context exemplars are incorporated directly into prompts, aligning outputs with the ontology. Moreover, extraction is designed as an iterative process: if the downstream KG Evaluation Agent detects low-confidence triples or cross-section inconsistencies, it returns structured feedback that is injected into the extractor's next prompt. This establishes a controllable, self-correcting feedback loop that progressively guides the agent toward higher-quality outputs.

Formally, given a document $D$ segmented into canonical sections $\{S_1, \ldots, S_m\}$, the agent applies $\Phi_k$ to each $S_k$, producing candidate triples of the form $(h, r, t, \xi)$, where $h$ and $t$ denote entities, $r$ is an ontology-aligned relation type, and $\xi$ is an evidence locator (e.g., character offsets or section identifiers). The global extracted graph is the union:

$$G_0 = \bigcup_{k=1}^{m} \{(h, r, t, \xi)\},$$

with triples serialized together with provenance metadata to support fine-grained auditing and downstream reasoning.

## 3.3 KG EVALUATION AGENT: MULTI-DIMENSIONAL SCORING AND ITERATIVE REFINEMENT

The KG Evaluation Agent serves as a graph-level auditor, providing rigorous quality control for the candidate graph $G$ generated by the extraction agent. This agent's role goes beyond verifying individual triples, as it is designed to capture global properties that simple checks cannot reveal.

To operationalize this audit, the evaluator performs a multi-dimensional assessment across five interpretable criteria: Domain Fit ($s_{\text{dom}}$), Accuracy ($s_{\text{acc}}$), Consistency ($s_{\text{con}}$), Completeness ($s_{\text{com}}$), and Granularity ($s_{\text{gra}}$). Each dimension is estimated via targeted LLM prompts (see Appendix A.5), producing a quality vector

$$\mathbf{q}(G) = (s_{\text{dom}},\, s_{\text{acc}},\, s_{\text{con}},\, s_{\text{com}},\, s_{\text{gra}}) \in [0,1]^5.$$

The overall graph quality is then defined as a weighted aggregation of these scores:

$$Q(G) = \sum_{i=1}^{5} w_i \cdot s_i, \qquad \sum_{i=1}^{5} w_i = 1,$$

where the weights $\{w_i\}$ are calibrated via ablation studies (§5), with higher emphasis placed on accuracy and consistency as they are most predictive of downstream QA performance. This transforms qualitative review into a principled and transparent quality signal.

Crucially, evaluation is not a static scoring step but a corrective supervision loop. When $Q(G) < \tau$, the evaluator generates actionable diagnostics—for instance, "low-confidence triples concentrated in *Methods*" or "entity *Model-A* inconsistently referenced across *Methods* and *Results*." These diagnostics are injected into the extractor's next-round prompts, converting a score into targeted guidance. Formally,

$$G^{(t+1)} = R\Big(G^{(t)}, \mathbf{q}(G^{(t)})\Big),$$

where $R(\cdot)$ denotes a refinement operator that integrates evaluator feedback into the section-aware, ontology-constrained extraction process. Iteration continues until $Q(G) \geq \tau$, yielding a finalized, auditable graph $G^{\star}$.

Through this design, the evaluator elevates extraction from a one-shot process into a robust, self-correcting pipeline. It not only detects deficiencies but prescribes remedies, ensuring that the resulting knowledge graph is reliable, verifiable, and a solid foundation for downstream QA generation.

### 3.4 QA GENERATION AGENT: SCHEMA-CONSTRAINED AND EVIDENCE-TRACEABLE QA

From the validated graph $G^*$, the QA Generation Agent identifies multi-hop reasoning paths that are explicitly traceable to textual evidence spans in the source document $D$. Formally, a reasoning chain is represented as

$$P = (e_1 \xrightarrow{r_1} e_2 \xrightarrow{r_2} \ldots \xrightarrow{r_k} e_{k+1}),$$

where $e_i$ are entities and $r_i$ are ontology-aligned relations. Each path $P$ is paired with its supporting evidence evidence$(P)$.

A schema-constrained LLM instantiation explicitly maps each reasoning path and its grounded evidence into a structured question–answer pair (prompts in Appendix A.6). Let $f_\theta$ denote the LLM, $\Psi(\cdot)$ a schema-constrained prompt template, and $\delta(\cdot)$ a constrained decoding policy (e.g., type/format validator). For any path $P \subseteq G^*$ with evidence $\Gamma(P)$, the QA pair is generated as:

$$(Q, A) = \delta\big(f_\theta\big(\Psi(P, \Gamma(P))\big)\big).$$

This formulation tightly couples questions, answers, supporting spans, and reasoning chains. The complete QA corpus is therefore defined as:

$$\mathcal{Q} = \big\{(Q, A, P, \Gamma(P)) \; : \; (Q, A) = \delta(f_\theta(\Psi(P, \Gamma(P)))), \; P \subseteq G^*\big\}.$$

Here $Q$ denotes the question, $A$ the answer, $P$ the reasoning path in the knowledge graph $G^*$, and $\Gamma(P)$ the evidence set grounded in the source text. The schema constraints in $\Psi$ and $\delta$ enforce type-correct entities and valid answer formatting, ensuring both structural validity and evidence traceability. By varying the hop length $k$ and relation types, the generator provides explicit control over reasoning difficulty.

In summary, the QA Generation Agent completes our closed-loop framework by integrating three key mechanisms: schema-driven constraints for structural validity, explicit path–evidence coupling for traceability, and exemplar conditioning for difficulty control. Together, these principles transform LLM capabilities into a systematic and reproducible pipeline for building high-quality academic QA resources.

Table 2: Evaluation of knowledge graph extraction quality, comparing zero-shot and LLM-based methods using manual annotated gold-set (Precision/Recall/F1) and *LLM-as-a-Judge* scores. All *LLM-as-a-Judge* scores are averaged across three LLM judges, and the *Final* column reports the average over evaluation dimensions.

| Category | Method | Gold-set Evaluation | | | LLM-as-a-Judge Evaluation | | | | | |
|---|---|---|---|---|---|---|---|---|---|---|
| | | Prec. | Rec. | F1 | Dom | Acc | Con | Com | Gra | Final |
| Zero-shot Extraction Methods | Self-Prompting | 35.7 | 35.7 | 35.7 | 4.9 | 5.0 | 5.2 | 5.1 | 5.6 | 5.2 |
| | QA4RE | 32.3 | 30.4 | 31.3 | 5.8 | 5.1 | 5.0 | 5.5 | 5.3 | 5.3 |
| | ChatIE | 16.5 | 34.1 | 21.4 | 5.9 | 5.3 | 5.2 | 5.6 | 5.3 | 5.3 |
| LLM-based Extraction Methods | Gemini-1.5-pro | 23.4 | 17.5 | 20.0 | 8.9 | 6.9 | 7.1 | 6.2 | 7.1 | 7.3 |
| | DeepSeek-V2.5 | 37.0 | 34.0 | 37.0 | 8.1 | 7.3 | 7.7 | 6.5 | 7.4 | 7.4 |
| | Claude-3.5 Opus | 30.0 | 28.6 | 29.3 | 9.1 | 7.0 | 7.4 | 6.7 | 7.2 | 7.5 |
| | o3 | 35.0 | 26.4 | 30.1 | 8.1 | 7.3 | 7.5 | 7.3 | 7.4 | 7.5 |
| | o1 | 53.8 | 26.4 | 35.4 | 9.3 | 8.1 | 8.8 | 6.8 | 7.7 | 8.1 |
| | Ours | **73.6** | **71.2** | **72.3** | **9.3** | **8.5** | **8.9** | **7.5** | **7.5** | **8.3** |

# 4 EXPERIMENTS

## 4.1 EXPERIMENTAL SETUP

**Datasets.** We construct a large-scale corpus by collecting scholarly articles from multiple open repositories[1][2]. In total, it comprises over 4,000 papers, from which our framework automatically generates more than 20,000 high-quality QA pairs, forming *AutoQG$^{20k}$*. Furthermore, we release the *AutoQG$^{7k}$* benchmark, a challenging subset designed with harder QA pairs that current powerful LLMs struggle to answer. Detailed statistics are provided in Table 1.

Table 1: Dataset statistics of *AutoQG$^{20k}$* and *AutoQG$^{7k}$*

| Metric | *AutoQG$^{20k}$* | *AutoQG$^{7k}$* |
|---|---|---|
| Papers | 4,435 | 4,435 |
| Triples | 243,214 | 243,214 |
| QA pairs | 20,424 | 7,921 |

**Evaluation.** To evaluate the effectiveness of our framework, we assess both the quality of the constructed knowledge graphs and the generated QA pairs. We adopt a dual evaluation strategy: (i) Human evaluation, where a subset of papers from our corpus is manually annotated under a double-blind protocol to provide gold standards; and (ii) LLM-as-a-Judge, where external LLMs are employed to provide multi-dimensional quality assessments. In addition, for QA generation evaluation, we also conduct human scoring to verify the reliability and fairness of the LLM-as-a-Judge results. Detailed evaluation criteria for KG quality and QA generation are presented in Sections 4.2 and 4.3.

**Implementation Details.** We implement our multi-agent pipeline using the LangGraph[3] framework, which models the workflow as a state machine with explicit routing and retry mechanisms. For generations, we primarily use *GPT-4o* and *GPT-4-turbo* with temperature fixed at 0.0 to ensure deterministic outputs.

## 4.2 MULTI-DIMENSIONAL EVALUATION ON KNOWLEDGE GRAPH EXTRACTION

**Baselines.** We evaluate the effectiveness of our section-aware and ontology-guided extraction strategy against two categories of representative baselines:

- *Zero-shot Baseline Methods* – including Self-Prompting, QA4RE, and ChatIE. These approaches serve as strong relation extraction baselines at the sentence- and document-level, operating in a zero-shot setting under predefined schemas.

---

[1] https://arxiv.org

[2] https://aclanthology.org

[3] https://www.langchain.com/langgraph

Table 3: Human evaluation and LLM-as-a-Judge evaluation results. All LLM-as-a-Judge scores are averaged across three independent judges. Scores are averaged across QA pairs and reported on a 0–10 scale.

| Method | Human Evaluation (0–10) | | | | | | LLM-as-a-Judge (0–10) | | | | | |
|---|---|---|---|---|---|---|---|---|---|---|---|---|
| | Rel | Ans | Distr | Flu | Reas | Evid | Rel | Ans | Distr | Flu | Reas | Evid |
| Gemini-1.5 Pro | 8.6 | 8.4 | 8.3 | 9.2 | 8.1 | 8.0 | 8.1 | 8.9 | 8.6 | 9.3 | 7.6 | 8.0 |
| DeepSeek-V2.5 | 8.8 | 8.5 | 8.8 | 9.0 | 7.9 | 8.2 | 8.9 | 9.0 | 8.3 | **9.5** | 8.0 | 8.0 |
| **Ours** | **9.8** | **9.8** | **8.9** | **9.4** | **8.4** | **9.7** | **9.6** | **9.6** | **9.0** | **9.5** | **8.5** | **9.6** |

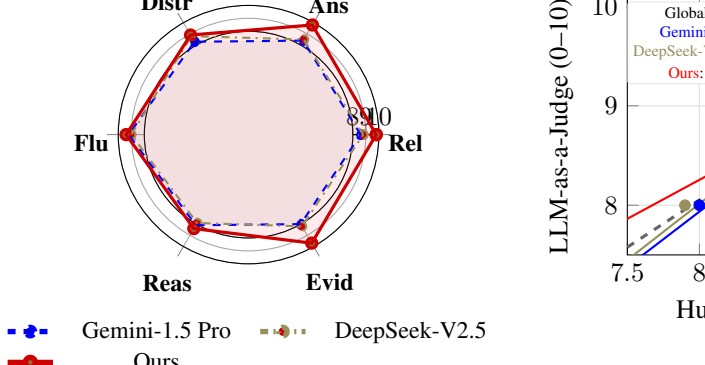

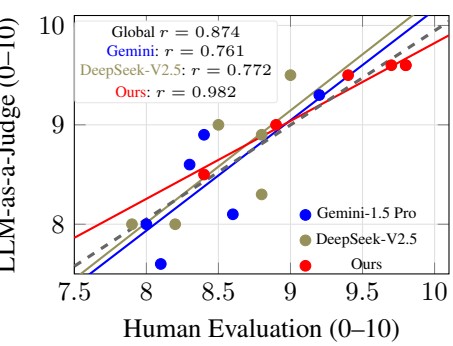

(a) Six-Dimension Radar Chart of human evaluation results

(b) Human vs. LLM-as-a-Judge (scatter + per-model fits + global fit)

Figure 3: Comparison of Human and LLM-as-a-Judge evaluations: (a) Radar chart across six dimensions; (b) Scatter plot with individual regression lines for each model and an overall (dashed) global regression.

- *LLM-based Extraction Methods* – including representative LLMs, which construct knowledge graphs directly from text by prompting with predefined schemas (the full schema is provided in Appendix A.10).

**Metrics.** We assess knowledge graph quality through both human and automated evaluation protocols. For human evaluation, we report precision, recall, and F1 scores on a gold dataset of 30 manually annotated papers. The distribution of annotators and detailed annotation guidelines are provided in Appendix A.2. For automated evaluation, we adopt an *LLM-as-a-Judge* protocol, where multiple strong LLMs (*GPT-5*, *Claude-3.5-Sonnet*, and *Qwen-2.5$^{72B}$*) independently score each extracted graph along five interpretable dimensions: *Dom: domain fit* (alignment with the target domain), *Acc: accuracy* (factual correctness of triples), *Con: consistency* (absence of contradictions across sections), *Com: completeness* (coverage of relevant relations), and *Gra: granularity* (level of detail captured).

As shown in Table 2, the results highlight: (i) **Human-Verifiable Precision.** Our framework achieves F1 = 72.3, far surpassing all baselines (best = 37.0), demonstrating accurate and complete extraction; (ii) **Multi-Dimensional Superiority.** It consistently outperforms across all five *LLM-as-a-Judge* dimensions, with the highest final score (8.3). These gains can be attributed to the integration of section-aware extraction, ontology-guided constraints, and iterative refinement. By leveraging discourse roles, our extractor reduces noise and captures domain-relevant triples with higher coverage; ontology alignment further suppresses spurious relations and enforces structural validity; and the evaluator's feedback loop ensures cross-sectional consistency and completeness.

### 4.3 QA GENERATION AND MULTI-DIMENSIONAL EVALUATION

We further assess the quality of multi-hop QA pairs generated by our framework.

**Baselines.** Given the fully automatic and annotation-free nature of our approach, we compare against strong LLM baselines, including *Gemini-1.5-Pro*, *Qwen2.5$^{72B}$*, and *DeepSeek-V2.5*.

Table 4: Extended evaluation of the generated QA benchmark under two settings: $AutoQG^{20k}$ (left block) and $AutoQG^{7k}$ (right block). Results are Strict Accuracy (%) on QA pairs, difficulty-controlled hops and accuracy *with Evidence*, where the evidence setting is evaluated on 3-hop questions.

| Category | Model | $AutoQG^{20k}$ | | | | $AutoQG^{7k}$ |
|----------|-------|------|------|------|---------------|------|
| | | 3-hop | 2-hop | 1-hop | with Evidence | 3-hop |
| *Base Models* | GPT-4 | 71.4 | 88.7 | 89.9 | 99.1 | 40.1 |
| | Llama-3.1[8B] | 65.5 | 78.2 | 83.0 | 91.3 | 36.3 |
| | Qwen-2.5[14B] | 67.3 | 81.3 | 86.0 | 99.7 | 30.6 |
| | DeepSeek-V2.5 | 64.7 | 82.7 | 86.9 | 98.6 | 33.7 |
| | Gemini-1.5 Flash | 65.8 | 77.3 | 87.5 | 98.7 | 29.0 |
| | Gemma-3[12B] | 73.2 | 84.0 | 95.2 | 92.6 | 33.2 |
| | Phi-2 | 71.0 | 81.2 | 89.5 | 98.0 | 45.4 |
| | WizardLM-2[70B] | 64.6 | 72.4 | 89.6 | 92.6 | 35.9 |
| *Reasoning Models* | Claude-3.5 Opus | 73.7 | 87.0 | 84.0 | 98.9 | 55.1 |
| | DeepSeek-R1 | 82.5 | 88.0 | 94.0 | 98.1 | 50.8 |
| | o1 | 82.3 | 84.0 | 95.0 | 98.2 | 50.0 |
| | o3 | 86.3 | 87.5 | 90.6 | 99.2 | 65.7 |

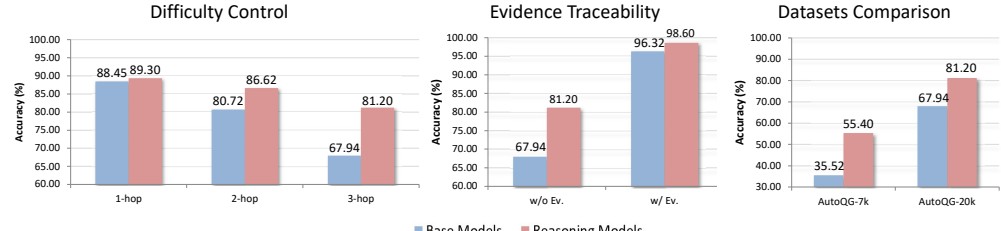

Figure 4: Comprehensive evaluation of the proposed benchmark on large language models. **Left:** performance trends under controlled reasoning difficulty by varying knowledge graph path length. **Center:** comparison of QA performance with and without evidence grounding. **Right:** results on $AutoQG^{7k}$ and $AutoQG^{20k}$.

**Metrics.** *Rel: relevance* (alignment of QA with context and evidence), *Ans: answerability* (whether the question can be correctly answered from the provided information), *Distr: distractor quality* (plausibility and challenge of incorrect options), *Flu: fluency* (clarity and grammatical correctness of the question and options), *Reas: reasoning depth* (extent to which multi-hop reasoning is required), and *Evid: Evidence Traceability: Explicit grounding in source text and path*. Furthermore, we conducted a rigorous double-blind human evaluation across six distinct dimensions. The evaluation was meticulously designed: all QA pairs were presented in a uniform style and randomized order.

As Table 3 shows, our framework achieves the best performance across all six dimensions, consistently surpassing strong LLM baselines. The largest gains are in relevance, answerability, and evidence, demonstrating that schema constraints and evidence grounding yield accurate and explicitly traceable QA pairs. Figure 3(a) further illustrates our method's higher scores and uniformly expanded radar shape, indicating balanced improvements. Figure 3(b) reveals a near-perfect correlation between human and LLM-as-a-Judge scores (r=0.982 for our method), confirming human-verifiable outputs.

# 5 A MULTI-PERSPECTIVE LLM ASSESSMENT WITH AUTOQG

To provide a holistic evaluation of our framework, we conduct a multi-perspective assessment of the constructed benchmark, focusing on three core dimensions: the reliability of difficulty control, the effectiveness of evidence traceability, and the robustness of LLM-based answering across diverse models.

Our framework regards reasoning depth by treating the length of knowledge graph paths as a proxy, and thus measures model performance on QA pairs derived from 1–3 hop reasoning chains within the same graph. For evidence traceability, we compare QA accuracy under two settings: with and without providing the corresponding reasoning path as explicit supporting evidence. Representative examples of QA pairs with varying path lengths and evidence availability are included in the Appendix A.3.

The extended evaluation in Table 7 and Figure 4 demonstrates the following key findings: (i) **Difficulty Control.** Reasoning path length directly correlates with task difficulty: as path length increases from 1-hop to 3-hop, model accuracy consistently decreases, confirming that path length is an effective proxy for reasoning depth; (ii) **Evidence Traceability.** Providing the reasoning path as supporting evidence substantially boosts accuracy, with most models approaching nearly 100% correctness. This highlights the strong evidence-grounded nature of our generated QA pairs; and (iii) **Model Robustness.** Advanced reasoning-oriented LLMs consistently outperform base models, indicating that our benchmark effectively distinguishes models by their reasoning capabilities rather than surface-level memorization. A representative example where most models fail is presented in Appendix A.8.

## 6 ABLATION STUDY

### 6.1 COMPONENT-WISE ABLATION ON AGENTS

We further conduct an ablation study to quantify the contribution of each design component.

**Metrics.** For the ablation study, we assessed the quality of the QA pairs generated using the same dimensions and the LLM-as-a-Judge protocol detailed in Section 4.3.

**Settings.** We remove one component at a time from different agents: (i) the *KG Generation Agent* by disabling Section-aware Prompting (SP) or Ontology Constraints (OC); (ii) the *KG Evaluation Agent* by removing the iterative feedback refinement loop (FB); and (iii) the *QA Generation Agent* by dropping question-format constraints (QF) or evidence grounding (E).

Table 5: Ablation study on the contribution of each module. Results are averaged across 3 LLM judges.

| Setting | Rel↑ | Ans↑ | Distr↑ | Flu↑ | Reas↑ | Evid↑ |
|---|---|---|---|---|---|---|
| **Ours** | **9.6** | **9.6** | **9.0** | **9.5** | **8.5** | **9.6** |
| **KG Generation Agent** | | | | | | |
| w/o SP | 9.2 | 9.0 | 7.0 | 9.3 | 5.5 | 8.8 |
| w/o OC | 8.8 | 8.5 | 6.0 | 9.0 | 5.0 | 8.9 |
| **KG Evaluation Agent** | | | | | | |
| w/o FB | 9.3 | 9.3 | 7.5 | 9.0 | 7.5 | 9.1 |
| **QA Generation Agent** | | | | | | |
| w/o QF | 9.0 | 9.0 | 7.2 | 9.2 | 6.1 | 8.9 |
| w/o E | 9.1 | 8.9 | 7.0 | 9.1 | 5.8 | 8.2 |

The ablation results in Table 5 highlight the complementary roles of all three agents in our framework. The *KG Generation Agent* is crucial for reasoning depth and distractor quality, as removing section-aware prompting (SP) or ontology constraints (OC) causes sharp drops in these dimensions. The *KG Evaluation Agent* mainly contributes to stability, with the absence of the feedback loop leading to noticeable declines in reasoning depth and coherence. The *QA Generation Agent* proves decisive for final QA quality: removing question-format constraints (QF) or evidence grounding (E) results in severe degradation of reasoning depth and evidence traceability. Overall, the strong performance of the full model stems from the synergy of these components, each indispensable for producing accurate, interpretable, and evidence-grounded QA pairs.

### 6.2 ABLATION STUDY ON INPUT MODALITIES

Beyond component-wise ablation, we further investigate how different input modalities contribute to QA generation. This study compares structured knowledge (KG) and unstructured source text (Sent) to examine whether relational structure or local linguistic grounding provides greater benefit.

**Settings.** We evaluate four input configurations: (i) KG + Sent (ours), which concatenates the KG reasoning path with the corresponding original text spans; (ii) Sent-only, which uses only the textual sentences containing KG node mentions without explicit KG structure; (iii) KG-only, which linearizes KG triples without textual spans; and (iv) Random-Sent, which samples length-matched but KG-irrelevant sentences from the same paper.

Table 6: Ablation on input modalities.

| Strategy | Rel | Ans | Distr | Flu | Reas | Evid |
|---|---|---|---|---|---|---|
| **S0 (KG + Sent) – Ours** | **9.6** | **9.6** | **9.0** | **9.5** | **8.5** | **9.6** |
| S1 (Sent-only) | 9.4 | 9.5 | 8.5 | 9.4 | 8.0 | 9.1 |
| S2 (KG-only) | 9.4 | 9.5 | 8.3 | 9.45 | 8.3 | 9.4 |
| S3 (Random-Sent) | 8.0 | 8.2 | 7.8 | 9.4 | 7.5 | 8.5 |

The comparison highlights the distinct roles of structural and textual inputs. The hybrid setting (S0) consistently yields the highest performance, demonstrating that KG and Sent are complementary rather than interchangeable. Compared with Sent-only (S1), adding KG structure substantially boosts **Reasoning Depth** and **Distractor Quality** ( +0.5), preventing the generation of shallow fact-retrieval questions. In contrast, compared with KG-only (S2), retaining the original text spans is critical for maintaining **Evidence Traceability** (avoiding a 0.2 drop) and **Fluency**, indicating that high-level relational reasoning must remain anchored to the linguistic context of the source paper.

Overall, these findings show that the KG provides a logical scaffold for compositional reasoning, while textual spans offer semantic grounding. Their combination is essential for generating high-quality, interpretable, and verifiable scientific QA pairs.

## 7 CONCLUSION

We introduced **AutoQG**, a multi-agent framework that transforms LLMs into a controllable pipeline for evidence-traceable QA generation. Applied to over 4,000 papers, it produced 243k triples and introduced the *AutoQG$^{20k}$* and *AutoQG$^{7k}$* benchmarks, explicitly linking more than 20,000 QA pairs to reasoning paths and evidence. Our framework outperformed strong baselines, demonstrating that ontology guidance and structured evaluation can reliably convert LLMs into pipelines that enhance precision, recall, and reasoning depth, providing a scalable benchmark for scientific QA. Nevertheless, challenges remain, including domain sensitivity, limited ontology coverage, and risks of bias propagation at scale. As future work, we will leverage the generated dataset to train a domain-adapted academic LLM, moving toward more robust, verifiable, and interpretable AI systems for scientific knowledge.

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

# A  APPENDIX

## A.1  USAGE OF LARGE LANGUAGE MODELS

The large language model (LLM) was employed solely for the purpose of polishing and refining the language of the manuscript. It assisted in improving grammatical accuracy, clarity, and overall readability. All intellectual content, ideas, and conclusions remain entirely those of the authors.

## A.2  KG ANNOTATION DETAILS

To establish the gold standard for our knowledge graph evaluation, we recruited seven master's students with strong backgrounds in natural language processing and machine learning to serve as annotators. Each annotator was assigned five papers and instructed to manually construct knowledge graphs in TTL format, following the predefined ontology detailed in Appendix A.4. The annotation process was overseen by a domain expert to ensure the quality and consistency of the output. The resulting manually annotated graphs served as the gold dataset for evaluating the performance of our KG Extraction Agent against various baselines.

## A.3  QA PAIR EXAMPLES

---

**Three-Hop QA Example**

**Reasoning Path:**

KISA $\xrightarrow{\text{outperforms}}$ SelfAttn $\xrightarrow{\text{evaluatedOn}}$ SemEval2010Task8 $\xrightarrow{\text{measures}}$ F1

**Context:**

- *"The integrated models improve performance by at least 0.9 F1 score and achieve new state-of-the-art results."*
- *"Besides TACRED, another dataset called SemEval2010-Task8 ... is used to evaluate the generalization ability."*
- *"We use the official macro-averaged F1 score as evaluation metric."*

**Question:**
Which model outperformed the Self-Attention model on the SemEval2010 Task 8 by utilizing which specific evaluation metric?

**Options:**

A. The KISA model outperformed the Self-Attention model using the official macro-averaged F1 score as the evaluation metric.

B. The KISA model surpassed the Self-Attention model using the average precision score on the SemEval2010 Task 8.

C. The Self-Attention model was outperformed by the KISA model, which used the micro-averaged recall score for evaluation.

D. The Self-Attention model was outperformed by the KISA model evaluated by the area under the ROC curve on SemEval2010 Task 8.

**Correct Answer:** A

**Explanation:**
The KISA model outperformed the Self-Attention model on SemEval2010 Task 8 using the official macro-averaged F1 score, as explicitly indicated by the reasoning path and evidence.

---

---

**Two-Hop QA Example**

**Reasoning Path:**

CameraSystemForFaceDetection $\xrightarrow{\text{uses}}$ RFIDEnergyHarvesting $\xrightarrow{\text{type}}$ Technology

**Context:**

• *"The camera system runs solely on energy harvested from RFID readers."*

• *"The camera system runs solely on energy harvested from RFID readers."*

**Question:**
What type of technology enables the camera system described in the paper to function solely on energy harvested from RFID readers?

**Options:**

A. RFID-based energy harvesting technology, which captures and converts energy from RFID systems.

B. Solar panel technology, which converts sunlight into electrical energy to power devices.

C. Piezoelectric energy harvesting technology, which generates power from mechanical stress.

D. Thermoelectric generators, which produce electricity from temperature differences.

**Correct Answer:** A

**Explanation:**
The camera system is explicitly described as operating with energy harvested from RFID readers. Therefore, the correct answer is RFID-based energy harvesting technology, consistent with the reasoning path and supporting evidence.

---

**One-Hop QA Example**

**Reasoning Path:**

LuisCeze $\xrightarrow{\text{type}}$ Author

**Context:**

• *"Amrita Mazumdar, Thierry Moreau, Armin Alaghi, Luis Ceze, Mark Oskin, Sung Kim, Meghan Cowan, Visvesh Sathe are mentioned as contributors."*

**Question:**
Which of the following accurately describes the professional role of Luis Ceze in the context provided in the paper?

**Options:**

A. Luis Ceze is mentioned as an author among other contributors in the paper.

B. Luis Ceze is recognized as the sole contributor and leading expert in the paper.

C. Luis Ceze is the primary editor responsible for reviewing and editing the paper.

D. Luis Ceze is a guest speaker who provided insights for the paper's development.

**Correct Answer:** A

**Explanation:**
Based on the knowledge graph path and context, Luis Ceze is listed along with other contributors as an author. No evidence supports alternative roles such as editor or guest speaker.

---

A.4   KG CONSTRUCTION AGENT PROMPTS

To ensure transparency and reproducibility, we provide the exact schema and prompt templates used in our framework. The schema defines the set of entity and relation types used to construct academic

knowledge graphs, while the prompts demonstrate how section-aware, ontology-guided extraction and evaluation are operationalized in practice. These resources allow others to replicate our pipeline and adapt it to new domains.

---

**Knowledge Graph Schema**

**Entity Types:**

- **Core Academic**: Models (GPT-4, BERT, Llama-2), Datasets (SQuAD, GLUE), Metrics (Accuracy, F1), Methods (Fine-tuning, LoRA), Algorithms (Attention, SGD), Architectures (Transformer).

- **Research Context**: Papers, Authors, Organizations (OpenAI, Google), Tools (PyTorch, TensorFlow), Venues (ACL, NeurIPS).

- **Measurement**: Scores, Parameters (7B, 175B), Configurations (hyperparameters, settings).

**Relation Types:**

- **Performance/Evaluation**: `:evaluatedOn`, `:achievesScore`, `:measures`, `:outperforms`, `:comparedWith`

- **Development/Creation**: `:basedOn`, `:improves`, `:proposes`, `:developedBy`, `:implementedIn`

- **Training/Application**: `:trainedOn`, `:finetunedOn`, `:uses`, `:appliedTo`, `:optimizedFor`

- **Research/Citation**: `:addresses`, `:publishedIn`, `:citedBy`

- **Structural**: `:partOf`, `:enabledBy`, `:resultIn`, `:relatedTo`

**Mandatory Metadata:**

`:contextText "original_text_snippet"`

---

**Knowledge Extraction Prompt**

**Role:** Expert extractor for academic knowledge. Output ontology-aligned triples for multi-hop reasoning and QA.

**Selective Extraction ($\leq 200$ Triplets):**

- Focus on major models, datasets, metrics, methods, and results.

- Extract only significant scores, parameters, comparisons, and settings.

- Skip minor mentions and redundant details.

**Triple Format:**

```
:Entity1 :relation :Entity2 ;
    :contextText "Original snippet..." .
```

**Examples:**

```
:GPT-4 :evaluatedOn :WinoGrande ;
    :contextText "GPT-4 was evaluated on WinoGrande dataset" .
:GPT-4 :achievesScore "94.2"^^xsd:float ;
    :contextText "GPT-4 achieved 94.2% accuracy" .
:ThisPaper :proposes :LogitLens ;
    :contextText "This paper proposes LogitLens" .
```

**Success Criteria:** 150–200 high-quality triples, schema-aligned, evidence-traceable, and relevant for QA.

---

## A.5 KG EVALUATION AGENT PROMPT

To complement the extraction schema, we also release the evaluation prompt used for graph-level quality assessment. The evaluator is designed to score an entire sectioned knowledge graph rather than individual triples, providing structured diagnostics and a weighted final score across five dimensions

(domain fit, accuracy, consistency, completeness, and granularity). This ensures that graph evaluation is transparent, reproducible, and aligned with downstream QA requirements.

---

**Sectioned KG Evaluator Prompt**

**Role.** You are an evaluator for a sectioned knowledge graph extracted from an academic paper. Score the *entire graph* (not individual triples) using **only** the given question, ontology, and sectioned evidence. Return **JSON only** (no extra text).

**Inputs**

- Task/Question: `{QUESTION}`

- Ontology & constraints (optional): `{ONTOLOGY_OR_RULES}`

- Sectioned KG snippet (RDF/Turtle-like): `{SECTIONED_TURTLE_BLOCK}`

*Notes:* Entities/relations may contain `:sourceSection`, `:sourceChunk`, and `:contextText`.

**What you must do**

- Parse the snippet into a graph (nodes/edges) with per-edge provenance `{section, chunk, evidence}`.

- Section-aware evidence: prefer evidence from the same `sourceSection`; if multiple, choose the strongest/clearest `contextText`.

- Compute graph-level diagnostics:

    - `triple_count`, `entity_count`, `section_coverage` (present / missing),
    - `conflict_count`, `missing_slot_rate`, `redundancy_rate`,
    - `granularity_notes`, `evidence_coverage` (portion of edges with `contextText`).

**Scoring dimensions (0–10) & section rules**

- **Domain Fit** – Alignment to task/question and paper domain across sections. Abstract/Conclusion weight high-level relevance; Methods/Results weight task-level relevance.

- **Accuracy** – Proportion of edges explicitly entailed by evidence; penalize speculation unless typical for the section (e.g., Discussion).

- **Consistency** – Cross-section coherence: contradictions, unit/time/version mismatches, ontology violations.

- **Completeness** – Coverage of section-appropriate slots/metadata; use `missing_slot_rate`.

- **Granularity** – Appropriate detail (normalized terms, alias resolution, versions, metric names/values); penalize systematically coarse/fine patterns per section norms.

$$\text{final\_score} = 0.20 \times \text{domain\_fit} + 0.20 \times \text{accuracy}$$
$$+ 0.20 \times \text{consistency} + 0.20 \times \text{completeness}$$
$$+ 0.20 \times \text{granularity}$$

(round to one decimal).

**Output (JSON only)**

Return a single JSON object. Keep reasons concise (each reason $\leq$ 30 chars; `summary_advice` $\leq$ 120 chars; each `top_fix` $\leq$ 60 chars).

```
{
  "meta": {
    "triple_count": <int>,
    "entity_count": <int>,
    "section_coverage": {
      "present": ["Abstract","Methods","..."],
      "missing": ["Results","..."]
    },
    "evidence_coverage": <0-1>,
    "conflict_count": <int>,
    "redundancy_rate": <0-1>,
```

```
      "missing_slot_rate": <0-1>,
      "granularity_notes": ["...","..."]
    },
    "scores": {
      "domain_fit":    {"score": <0-10>, "reason": "$\leq$30 chars"},
      "accuracy":      {"score": <0-10>, "reason": "$\leq$30 chars"},
      "consistency":   {"score": <0-10>, "reason": "$\leq$30 chars"},
      "completeness":  {"score": <0-10>, "reason": "$\leq$30 chars"},
      "granularity":   {"score": <0-10>, "reason": "$\leq$30 chars"}
    },
    "final_score": <0-10 one-decimal>,
    "summary_advice": "$\leq$ 120 chars, prioritized",
    "top_fixes": [
      "Fix 1 ($\leq$ 60 chars)",
      "Fix 2 ($\leq$ 60 chars)",
      "Fix 3 ($\leq$ 60 chars)"
    ]
  }
```

A.6 QA GENERATION AGENT PROMPT

---

**QA Generation Prompt (System)**

**Role:** Expert generator of educational multiple-choice questions grounded in academic paper knowledge graphs.

**Task:** Produce *one* high-quality MCQ with four options (A–D) from a **3-hop** KG reasoning path anchored to a specific paper section.

**Constraints:**

- Require multi-hop reasoning that follows the provided path end-to-end.

- Create exactly four options with *only one* correct answer.

- Keep options plausible yet clearly distinguishable; include mildly misleading overlaps.

- Use an academic, precise, objective style in **English**.

- Ground the question in the given section context; ensure it is answerable from the information provided.

- Prefer phenomena that deviate from common knowledge and are surprising.

- You may use a term a domain expert would understand but that is likely ambiguous for an LLM; this term must *not* appear elsewhere in the text.

- Each option must contain **more than 10 words**.

- Integrate *internal–external knowledge*: incorporate internal domain knowledge not verbatim in the paper while keeping the answer derivable from the provided path and context.

- Make the item challenging but resolvable without external resources beyond the provided inputs.

**Output Format (JSON only):**

```
{
  "question": "Clear, specific question text",
  "options": {
    "A": "First option (must be longer than 10 words)",
    "B": "Second option (must be longer than 10 words)",
    "C": "Third option (must be longer than 10 words)",
    "D": "Fourth option (must be longer than 10 words)"
  },
  "correct_answer": "A/B/C/D",
  "explanation": "Brief explanation of why the answer is correct"
}
```

**Success Criteria:** One valid-JSON output; reasoning explicitly implied by the 3-hop path; options long, plausible, and non-ambiguous; single correct answer with a concise justification.

---

**QA Generation Prompt (User)**

**Inputs:** Section name, knowledge path, and supporting context from the paper.

**Instruction:** Based on the following knowledge graph path and the accompanying section-level context, generate *one* multiple-choice question that requires reasoning through the entire **3-hop** path. The item should test understanding of entities and relations involved.

**Template (model receives the following fields):**

```
KNOWLEDGE PATH:
{path_description}=
CONTEXT FROM PAPER:
{context_text}
SECTION: {section}
```

**Return:** Respond with *valid JSON only* following the schema in the System prompt.

---

## A.7 MORE EVALUATION RESULTS ON LLMS

To provide a more comprehensive view of our benchmark, we extend the main evaluation by reporting additional results on a broader set of large language models in the Appendix. Consistent with the main paper, our analysis emphasizes three complementary dimensions: (i) **Difficulty Control**, measured by model performance across 1–3 hop reasoning paths; (ii) **Evidence Traceability**, quantified by accuracy with and without providing supporting paths; and (iii) **Model Robustness**, evaluated through comparisons between reasoning-oriented and base LLMs, as shown in Table 7.

Table 7: Extended evaluation of the generated QA benchmark under two settings: $AutoQG^{20k}$ (left block) and $AutoQG^{7k}$ (right, QA-Pair only). Results are Strict Accuracy (%) on QA pairs, difficulty-controlled hops, and accuracy with/without evidence.

| Category | Model | $AutoQG^{20k}$ | | | | $AutoQG^{7k}$ |
|----------|-------|------|-------|-------|-------------|---------|
| | | 1-hop | 2-hop | 3-hop | Acc (w/ Ev.) | QA-Pair |
| *Base Models* | GPT-4o | 95.2 | 91.0 | 72.5 | 96.1 | 43.3 |
| | Mistral Large 2 | 94.2 | 91.0 | 76.4 | 95.2 | 55.3 |
| | WizardLM-2$^{70B}$ | 89.6 | 72.4 | 64.6 | 92.6 | 35.9 |
| | Qwen-2.5$^{14B}$ | 86.0 | 81.3 | 67.3 | 99.7 | 30.6 |
| | DeepSeek-V2.5 | 86.9 | 82.7 | 64.7 | 98.6 | 33.7 |
| | Gimini-1.5-flash | 87.5 | 77.3 | 65.8 | 98.7 | 29.0 |
| | GLM-4 | 83.1 | 75.0 | 65.6 | 98.7 | 29.0 |
| | Claude-haiku | 82.9 | 79.7 | 60.2 | 99.7 | 31.4 |
| | Mistral-7B (Base) | 85.7 | 82.0 | 62.2 | 99.2 | 28.0 |
| | DBRX | 83.4 | 79.2 | 56.2 | 90.8 | 30.7 |
| *Reasoning Models* | Claude-3.5 Sonnet | 86.0 | 84.0 | 75.8 | 95.9 | 52.9 |
| | Grok-2 | 94.0 | 88.0 | 69.6 | 92.9 | 41.9 |
| | Mistral-7B | 76.5 | 69.2 | 60.7 | 90.9 | 27.6 |

## A.8 CASE STUDY ON LLM FAILURE CASES

---

**Example QA Pair**

**Question:**
In the study described in the *researchcontextandstudydesign* section, how was the approach for balancing functional requirements and non-functional requirements (NFRs) presented to the experts?

**Options:**

    A. The approach was embedded within the closed questions of the questionnaire on a 4-point Likert scale.

    B. Experts were invited to a workshop where the approach was discussed in detail before starting the questionnaire.

    C. It was detailed at the beginning of the study to introduce experts to the main focus of research.

    D. It was presented in the third part of a questionnaire that included both open and closed questions.

**Correct Answer:** D

**Reasoning Path:**
Questionnaire → Open and Closed Questions → Three Parts → Approach for Balancing Functional and NFRs

**Context:**
"The questionnaire comprised 20 open questions and 4 closed questions... Finally, in the third part we presented the experts our approach for balancing functional and NFRs."

---

**Case Study on Incorrect Answers by LLMs.** We analyze a representative failure case in which most LLMs struggle to answer correctly. This example requires a three-hop reasoning chain that links the questionnaire to its division into three parts and finally to the presentation of the proposed approach, yet many models fail to follow this path and instead latch onto superficial cues. Distractor options are particularly challenging, as references to Likert scales or common research practices (e.g., introducing the method at the beginning) appear plausible and mislead models into incorrect selections. Success thus hinges on capturing the precise detail that the approach was presented in the *third part* of the questionnaire, which most models overlook. We also note that inconsistencies in dataset annotations (e.g., mismatched explanations and correct answers) can exacerbate model errors. This case highlights that our benchmark effectively probes weaknesses in multi-hop reasoning, evidence grounding, and robustness against distractors.

## A.9 QA HUMAN EVALUATION DETAILS

We conducted a double-blind human evaluation to rigorously assess the quality of the generated QA pairs. The evaluation was performed by a separate group of annotators who were blind to the source of each QA pair. We collected QA pairs from multiple sources, including our framework and various baselines, and presented them to the annotators in a randomized, interleaved fashion. This approach prevented annotators from inferring the source and ensured an unbiased assessment. The annotators were provided with a detailed scoring rubric to evaluate each QA pair across six key dimensions: Relevance, Answerability, Distractor Quality, Fluency, Reasoning Depth, and Evidence Traceability. The scoring criteria for each dimension are summarized in the table 8.

| Dimension | Definition | Scoring Guideline (0–10, with Example) |
|---|---|---|
| Relevance | Alignment of QA with context, path, and evidence. | 0: Irrelevant; 10: Fully relevant. High score if QA targets core methods/results, not background. |
| Answerability | Answer can be derived from provided context/path. | 0: Unanswerable; 10: Clearly answerable. Penalize if external knowledge is needed. |
| Distractor Quality | Plausibility and misleadingness of distractors. | 0: Trivial/repetitive; 10: Plausible and non-trivial. |
| Fluency | Clarity and naturalness of language. | 0: Unclear; 10: Fluent and academic. High score if concise and grammatically correct. |
| Reasoning Depth | Need for multi-hop reasoning/evidence integration. | 0: Fact recall; 10: Multi-hop reasoning. Reflects difficulty control. |
| Evidence Traceability | Explicit grounding in source text and path. | 0: No link; 10: Explicitly linked. High score if tied to specific findings/experiments. |

Table 8: Evaluation dimensions for QA pairs with scoring rubric (0–10 scale).

## A.10 The prompt of KG Construction of Representative LLMs for KG evaluation

---

**Knowledge Graph Schema**

**Entity Types:**

- **Core Academic**: Models (GPT-4, BERT, Llama-2, Transformer), Datasets (WinoGrande, SQuAD, GLUE), Metrics (Accuracy, F1, BLEU, ROUGE, Perplexity), Methods (Fine-tuning, LoRA, Pre-training), Algorithms (Attention, Backpropagation, SGD), Architectures (Encoder-Decoder, Attention-only).

- **Research Context**: Papers, Authors, Organizations (OpenAI, Google, Meta), Tools (PyTorch, TensorFlow, Hugging Face), Venues (ACL, NeurIPS, journals).

- **Measurement**: Scores, Parameters (7B, 175B), Configurations (hyperparameters, experimental settings).

**Relation Types:**

- **Performance & Evaluation**: `:evaluatedOn`, `:achievesScore`, `:measures`, `:outperforms`, `:comparedWith`

- **Development & Creation**: `:basedOn`, `:improves`, `:proposes`, `:developedBy`, `:implementedIn`

- **Training & Application**: `:trainedOn`, `:finetunedOn`, `:uses`, `:appliedTo`, `:optimizedFor`

- **Research & Citation**: `:addresses`, `:publishedIn`, `:citedBy`

- **Structural**: `:partOf`, `:enabledBy`, `:resultIn`, `:relatedTo`

**Mandatory Metadata:**

`:contextText "original_text_snippet"`

---

## A.11 ERROR ANALYSIS AND REPRESENTATIVE CASES

To understand the typical failure modes of our KG extraction module, we conducted a detailed qualitative error analysis on a manually annotated subset consisting of 42 incorrect triples. The errors fall into three major categories: (1) incorrect relations, (2) over-extraction, and (3) under-extraction. Table 9 summarizes representative examples, along with their error mechanisms.

Table 9: Summary of error categories and mechanisms.

| Error Type | Ratio | Gold | Prediction | Mechanism |
|---|---|---|---|---|
| Incorrect Relation | 57.1% | `basedOn` | `uses` | Verb-pattern drift; misunderstanding methodological vs. implementation semantics |
| Incorrect Relation (Causalization Bias) | — | `relatedTo` | `arisesFrom` | Over-inference of causality; semantic over-mapping |
| Semantic Reversal | — | `appliedTo` | `retrievesFrom` | Directionality confusion; role inversion |
| Granularity Mismatch (Over-extraction) | 38.1% | `basedOn` | `uses, implements` | Predicting multiple fine-grained relations instead of one abstract relation |
| Context Misalignment | — | `basedOn` | `uses` | Confusion between hyperparameter usage and algorithmic design dependency |
| Entity-type Mismatch | — | `extends` | `improvesOver` | Cross-level abstraction drift; technical hierarchy confusion |
| Entity-type Mismatch (Resource-level) | — | `uses` | `usesKG` | Over-specific inference at resource level |
| Under-extraction | 4.8% | `missing` | — | Long-context cues not captured; discourse-level limitations |

The majority of failures arise from *verb-pattern drift* and *semantic over-mapping*, where surface-level verb cues (e.g., "use", "incorporates", "arises from") incorrectly trigger fine-grained or causally loaded relations. A second major error source is *granularity mismatch*: the model often produces several fine-grained relations instead of one abstract conceptual relation (e.g., predicting both `uses` and `implements` instead of the gold `basedOn`). Finally, we observe *entity-type mismatches* that reveal cross-level abstraction inconsistencies (e.g., model-level vs. concept-level), as well as occasional under-extraction due to long-context dependency failures.

These patterns align with our earlier findings and directly motivate the improvement strategies proposed in the main text, particularly the need for contextual verb–relation disambiguation, hierarchical relational modeling, and enhanced discourse-level understanding.

## A.12 REASONING PATH STRATEGY.

To ensure that the reasoning chains used for QA generation are meaningful rather than arbitrary, the QA Generation Agent does **not** sample reasoning paths randomly. Instead, we employ a *structure-constrained traversal* strategy that enforces semantic coherence across sections while preventing trivial or circular reasoning. To quantify the contribution of this design, we compare three traversal strategies as follows:

To evaluate the impact of the traversal strategy, we follow the six-dimensional LLM-as-a-Judge protocol described in Section 4.3 and report averaged scores on AutoQG20k:

Across all dimensions, the structure-constrained traversal (P0) yields the strongest results, with the most pronounced gain observed in the **Reasoning** dimension (8.5 vs. 7.9/8.2). This indicates

| Strategy | Hop Constraints | Rationale & Design Logic |
|---|---|---|
| **P0: AutoQG (Ours)** | $\leq 3$ hops; acyclic; cross-sectional; prefers 2–3 hops | Encourages semantic information synthesis across different sections (e.g., Method $\rightarrow$ Result) and suppresses logically vacuous loops |
| **P1: Pure Random Walk** | Random walk, hops $\in \{1, \ldots, 5\}$ | Represents unconstrained exploration over shallow and deep paths |
| **P2: Length-fixed Random** | Random walk, $\leq 3$ hops | Controls for hop length to isolate whether structural constraints (rather than length) yield performance gains |

| Strategy | Rel | Ans | Distr | Flu | Reas | Evid |
|---|---|---|---|---|---|---|
| **P0: AutoQG (Ours)** | **9.6** | **9.6** | **9.0** | **9.5** | **8.5** | **9.6** |
| P1: Random Paths | 9.5 | 9.0 | 8.6 | 9.4 | 7.9 | 9.1 |
| P2: Length-Constrained Random | 9.5 | 9.0 | 8.6 | **9.5** | 8.2 | 9.3 |

that enforcing *cross-sectional and acyclic* paths prevents purely local or shallow reasoning and encourages the model to integrate heterogeneous evidence. The same constraint also leads to higher **Answerability** and **Evidence Traceability**, suggesting that meaningful path design directly improves QA fidelity rather than simply increasing hop count.

## A.13 ADDITIONAL RESULTS ON QA GENERATION

To further ensure that our conclusions remain robust against the newest generation of large language models, we extend our evaluation to a broader set of *frontier models*, including the most recent GPT and Gemini variants, as well as competitive open-source models. All models were evaluated under the same LLM-as-a-Judge protocol used in Section 4.3, ensuring fair and fully comparable scoring across dimensions.

Table 10: Comparison with newly released frontier LLMs on QA generation quality. Our method maintains a consistent lead across all evaluation dimensions.

| Model | Rel | Ans | Distr | Flu | Reas | Evid |
|---|---|---|---|---|---|---|
| gpt-5 | 7.4 | 7.2 | 7.4 | 8.6 | 6.8 | 6.9 |
| gpt-5-mini | 8.0 | 7.9 | 8.3 | 9.0 | 7.5 | 7.7 |
| gpt-5.1 | 9.4 | 9.0 | **9.2** | 9.4 | 8.6 | 8.4 |
| claude-sonnet-4-thinking-all | 8.4 | 7.9 | 8.7 | 9.3 | 7.6 | 7.6 |
| Baichuan4 | 8.9 | 8.6 | 9.0 | 9.4 | 8.0 | 7.9 |
| Qwen_Qwen3-32B | 8.2 | 8.0 | 8.0 | 8.9 | 7.4 | 7.5 |
| THUDM_GLM-4-32B-0414 | 8.6 | 8.3 | 8.7 | 9.3 | 7.8 | 7.8 |
| THUDM_GLM-4-9B-0414 | 7.7 | 7.5 | 7.6 | 8.8 | 7.1 | 7.1 |
| doubao-seed-1-6-thinking-250715 | 8.4 | 8.0 | 8.4 | 9.2 | 7.6 | 7.6 |
| gemini-2.5-pro-exp-03-25-thinking | 8.6 | 8.4 | 8.7 | 9.3 | 7.8 | 7.9 |
| glm-4 | 7.6 | 7.4 | 7.5 | 8.6 | 7.0 | 7.0 |
| llama-3.1-70b-instruct | 7.0 | 6.9 | 7.2 | 8.2 | 6.6 | 6.6 |
| **Ours** | **9.6** | **9.6** | 9.0 | **9.5** | **8.5** | **9.6** |

The expanded comparison firmly supports the findings reported in the main paper. Our method consistently maintains the highest performance across all key metrics, even when evaluated against the most recent frontier systems such as GPT-5.1 and Gemini-2.5-Pro-Thinking. The stability of relative model rankings across strong LLMs further demonstrates the robustness and discriminatory power of our evaluation setup, confirming that the advantages of our framework are not tied to older model versions but remain valid against the current state of the art.

## A.14 FREE-FORM QUESTION GENERATION USING AUTOQG

Although the main paper focuses on multiple-choice questions (MCQs) to enable deterministic and scalable evaluation, the proposed *Path→Question* generation mechanism is inherently format-agnostic. To verify the generalizability of AutoQG beyond MCQs, we conducted an additional study in which the QA pipeline was used to generate *free-form short-answer* questions using the same KG-guided reasoning process.

**Evaluation Dimensions.** To ensure a robust assessment of short-answer QA quality, we designed a six-dimensional evaluation protocol shown in Table 11, capturing both linguistic properties and reasoning/evidence requirements.

Table 11: Evaluation dimensions for free-form QA generation.

| Dimension | Abbreviation | Description |
|---|---|---|
| Relevance | Rel | Degree to which the question is grounded in the given KG path and supporting section context. |
| Answer Quality | AnsQ | Correctness and completeness of the reference answer with respect to the intended semantics of the question. |
| Factual Correctness | Fact | Faithfulness of all factual statements in the answer relative to the provided evidence, without hallucination or unsupported inference. |
| Clarity & Fluency | Flu | Linguistic clarity, phrasing naturalness, and structural coherence of both question and answer. |
| Reasoning Depth | Reas | Degree to which answering the question requires multi-step reasoning or integration of distributed information rather than direct lookup. |
| Evidence Traceability | Evid | Extent to which the answer can be explicitly supported by evidence from the KG path or context span. |

**Experimental Results.** The scoring results for free-form QA generation are shown in Table 12.

Table 12: LLM-as-a-Judge results for free-form QA generation.

| Rel | AnsQ | Fact | Flu | Reas | Evid |
|---|---|---|---|---|---|
| 9.6 | 9.5 | 9.5 | 9.5 | 8.5 | 9.6 |

These findings confirm that AutoQG extends robustly to open-ended short-answer formats. The high **Factual Correctness** score (9.5) highlights the low hallucination rate, attributable to the KG path acting as an explicit and verifiable content plan guiding answer construction. **Evidence Traceability** remains strong (9.6) even without the constrained answer space of MCQs, indicating that the KG structure continues to enforce transparency and verifiability in free-form generation.

## A.15 COST ANALYSIS OF THE AUTOQG MULTI-AGENT PIPELINE

This section presents additional analysis of the computational cost of the AutoQG multi-agent pipeline. Although AutoQG coordinates multiple specialized agents, the decomposition substantially improves efficiency by reducing the token length per call and distributing reasoning across smaller modules, preventing the exponential scaling commonly observed in long-context end-to-end prompting.

**Multi-Agent Cost Efficiency.** Table 13 compares AutoQG against a single-call end-to-end prompting baseline. While AutoQG triggers more calls per paper, its modularization dramatically reduces the average token count per call while yielding significantly higher QA quality.

Table 13: Cost characteristics and QA performance: End-to-end prompting vs. AutoQG.

| Setting | Avg. Calls / Paper | Avg. Tokens / Call | QA Quality | Rel | Ans | Distr | Reas |
|---|---|---|---|---|---|---|---|
| End-to-end LLM QA | 13,421.61 | 13,421.61 | 7.31 | 8.0 | 8.9 | 7.4 | 2.4 |
| **AutoQG (multi-agent)** | 23,592.53 | **4,831.52** | **8.45** | **9.6** | **9.6** | **9.1** | **8.4** |

The results show that multi-agent decomposition functions as a cost-control mechanism. By bounding sequence length and separating extraction, evaluation, and question formation, AutoQG yields a +1.14 improvement in overall QA quality and a +6.0 improvement in reasoning depth while keeping token consumption manageable and predictable.

### A.16 REPRODUCIBILITY AND OPEN-SOURCE DEPLOYMENT

AutoQG is not structurally tied to closed-source LLMs. Because each agent functions independently, fine-tuning or replacing any component does not affect the others. To examine practical portability, we re-executed the full workflow using compact open-source models. The KG Extraction Agent and QA Generation Agent used `Qwen-7B-Instruct`, and the KG Evaluation Agent used `DeepSeek-V2.5`.

Table 14: KG evaluation performance using open-source models within AutoQG.

| Model | Dom | Acc | Con | Com | Gra | Overall QG |
|---|---|---|---|---|---|---|
| AutoQG (Qwen-7B-Instruct) | 7.0 | 7.7 | 7.0 | 6.0 | 6.3 | 7.0 |
| AutoQG (DeepSeek) | 9.0 | 8.3 | 8.3 | 7.0 | 7.3 | 8.0 |

Table 15: QA generation performance using open-source models within AutoQG.

| Model | Rel | Ans | Distr | Flu | Reas | Evid |
|---|---|---|---|---|---|---|
| AutoQG (Qwen-7B-Instruct) | 8.6 | 8.3 | 7.7 | 9.0 | 7.4 | 8.3 |
| AutoQG (DeepSeek) | 9.4 | 9.1 | 8.5 | 9.5 | 8.4 | 9.2 |

**Final Cost Estimate.** When deployed with DeepSeek-based agents, the full pipeline (extraction $\rightarrow$ evaluation $\rightarrow$ QA generation) averages **\$0.06 per paper**. The modular architecture therefore ensures both (i) high fidelity and (ii) low-cost reproducibility: computational cost grows linearly rather than exponentially with task complexity, and each agent can be fine-tuned or replaced independently for fully open-source deployment.

