# OpenReview forum: "AutoQG: An Automated Framework for Evidence-Traceable Question Generation via Ontology-Guided Knowledge Graph Construction"
_ICLR.cc/2026/Conference — ICLR 2026 Conference Withdrawn Submission_

### Official Review · Reviewer_hCUA · 2025-11-01

**Soundness:** 3
**Presentation:** 3
**Contribution:** 3
**Rating:** 4
**Confidence:** 3

**Summary:**

This paper introduces AutoQG, a fully automated, multi-agent framework that generates evidence-traceable, scientific question-answer (QA) pairs from academic papers without human annotation. It addresses the key challenges of existing methods: costly manual annotation and the generation of unverifiable outputs by LLMs.

AutoQG's pipeline consists of three agents:

* KG Extraction Agent: Uses section-aware and ontology-guided prompts to build a structured KG from a paper's text.

* KG Evaluation Agent: Performs a multi-dimensional evaluation of the KG to ensure its accuracy and consistency, providing feedback in an iterative refinement loop.

* QA Generation Agent: Uses the validated KG to produce schema-constrained QA pairs that are explicitly grounded in the graph's reasoning paths and the original source evidence.

**Strengths:**

* The agents' tasks are simple but intelligently implemented, providing a useful, scalable framework to explore potentially for other non QA generation domains.

* The auditability (as the KG provides an "auditable backbone" for the generation process), traceability (it allows for the generation of QA pairs that are "explicitly traceable to the source text"), and the ability to create questions with multihop reasoning of the method.

**Weaknesses:**

* In Table 3, you compare against now quite outdated frontier models. How would the table look with newly released models like GPT-5?

* Given the multiagent setup, you do not address the increase in cost vs a standard LLM call.

**Questions:**

* While this appears to be a useful, unique QA generation method, it is often the follow-up questions (and follow-ups to those follow-ups etc) that are more akin to real-life science work. Could you extend your method to evaluate this in any way?

* Can you share the more granular details of what types of questions were judged highly by humans or not in Table 3 across all methods?

---

> ### Author Response · Authors · 2025-11-22
> **Response to Reviewer hCUA Part I**
>
> > How would the results look with newly released models like GPT-5?
>
> **R1: Frontier Model Comparison**
>
> We are grateful to the reviewer for highlighting the critical issue of model freshness and the perceived limits of our comparison set. In response to this insightful critique, we conducted a complete evaluation using the newest generation of frontier models, including $\text{GPT-5}$, $\text{GPT-5.1}$, $\text{Gemini-2.5-Pro-Thinking}$, and others. We added more model comparison in our manuscript in **section A.13**.
>
> | **Model**                         | **Rel** | **Ans** | **Distr** | **Flu** | **Reas** | **Evid** |
> |-----------------------------------|---------|---------|-----------|---------|----------|----------|
> | gpt-5                             | 7.4     | 7.2     | 7.4       | 8.6     | 6.8      | 6.9      |
> | gpt-5-mini                        | 8.0     | 7.9     | 8.3       | 9.0     | 7.5      | 7.7      |
> | gpt-5.1                           | 9.4     | 9.0     | **9.2**       | 9.4     | 8.6      | 8.4      |
> | claude-sonnet-4-thinking-all      | 8.4     | 7.9     | 8.7       | 9.3     | 7.6      | 7.6      |
> | Baichuan4                         | 8.9     | 8.6     | 9.0       | 9.4     | 8.0      | 7.9      |
> | Qwen_Qwen3-32B                    | 8.2     | 8.0     | 8.0       | 8.9     | 7.4      | 7.5      |
> | THUDM_GLM-4-32B-0414              | 8.6     | 8.3     | 8.7       | 9.3     | 7.8      | 7.8      |
> | THUDM_GLM-4-9B-0414               | 7.7     | 7.5     | 7.6       | 8.8     | 7.1      | 7.1      |
> | doubao-seed-1-6-thinking-250715   | 8.4     | 8.0     | 8.4       | 9.2     | 7.6      | 7.6      |
> | gemini-2.5-pro-exp-03-25-thinking | 8.6     | 8.4     | 8.7       | 9.3     | 7.8      | 7.9      |
> | glm-4                             | 7.6     | 7.4     | 7.5       | 8.6     | 7.0      | 7.0      |
> | llama-3.1-70b-instruct            | 7.0     | 6.9     | 7.2       | 8.2     | 6.6      | 6.6      |
> | **Ours**                              | **9.6**     |  **9.6**    | 9.0         |  **9.5**    |  **8.5**     |  **9.6**     |
>
> As shown in the table above, the expanded analysis decisively confirms and strengthens our original findings. Our proposed method **maintains a competitive performance advantage across all key metrics compared to the newly evaluated frontier models** . We sincerely appreciate this suggestion, which resulted in a more comprehensive and strongly supported experimental analysis for our paper.
>
> > Could the method be extended to generate follow-up questions?
>
> **R2: Extension of AutoQG to Multi-step Follow-up Question Generation**
>
> We thank the reviewer for raising this insightful question. We fully agree that real scientific inquiry is inherently multi-turn, involving iterative follow-up questions that drill down from concepts to specific details. We confirm that AutoQG naturally supports this extension without requiring modification to any core module; it only leverages the structural priors already encoded in the Academic KG.The key idea is to **operate on localized knowledge structures**. We extend the single-turn approach into a coherent multi-turn dialogue by following a structured procedure:
>
>  - Local Knowledge Tree Construction: Given a central scientific concept (e.g., a Method or Model), we use the existing KG and ontology constraints (**Appendix A.10**) to construct a local KG subtree of depth $\geq 3$ rooted at that concept.
>
>  - Scientific Inquiry Path Selection: We select a single root-to-leaf path from this tree as the "follow-up reasoning chain." This path is chosen to follow a logical progression, enforced by the ontology (e.g., $\text{Method} \rightarrow \text{Mechanism} \rightarrow \text{Experiment} \rightarrow \text{Limitation}$).
>
>  - Multi-Turn Generation: The existing QA Generation Agent is used iteratively along the path. By constraining the LLM to the evidence of the next hop, we ensure:
>    - Progressive Depth: Each turn deepens the inquiry (e.g., Turn 1 asks for the overview, Turn 2 asks for the specific component).
>    - Evidence Continuity: Every question is explicitly grounded in both the KG relations and the exact evidence sentences of the corresponding hop.
>
> To illustrate the robust structural extensibility of this AutoQG-Dialogue mechanism, we provide a small, concrete demonstration:
>  - We extracted a 4-layer local KG subtree: https://anonymous.4open.science/r/ICLR-Supplementary-Material-F488-13640/Tree-Case.png
>  - We generated the corresponding multi-turn follow-up sequence: https://anonymous.4open.science/r/ICLR-Supplementary-Material-F488-13640/multi-turn-case.json
> This example clearly shows a coherent 4-turn question chain that mirrors scientific questioning, while proving that the evidence traceability and structural integrity of our single-turn AutoQG can be seamlessly extended to multi-step scientific inquiry, which we identify as a key direction for future work.

---

> ### Author Response · Authors · 2025-11-22
> **Response to Reviewer hCUA Part II**
>
> > Can you share more granular details on which types of questions were scored highly or poorly by human evaluators in Table 3 across methods?
>
> **R3: Granular Analysis of QA Quality**
>
> We appreciate the opportunity to provide granular details. A qualitative breakdown of QA samples confirms that human ratings are strongly correlated with the locality of evidence and the explicitness of reasoning chains. Our benchmark effectively distinguishes between two types of questions:
>
> | Question Type | High-Scoring QA (Score 9–10) | Low-Scoring QA (Score <8) |
> | --- | --- | --- |
> | **Evidence Locality** | Requires explicit recall or direct causal links (e.g., “Algorithm A uses Metric B”). | Requires implicit synthesis (“Inference Gap”) that goes beyond explicitly stated evidence. |
> | **Reasoning Demand** | Low “Inference Gap”; reasoning path is simple and incontrovertible. | High inference required, often demanding deduction of implied (not stated) limitations. |
> | **Evaluation Focus** | Excellent evidence traceability and semantic stability. | Poor reasoning depth and high distractor entanglement. |
>
> To be more detailed, High-scoring questions (examples available in https://anonymous.4open.science/r/ICLR-Supplementary-Material-F488-13640/High-Score-QA-Pairs) clearly validate the strength of our ontology-guided generation. These pairs exhibit high Semantic Stability by relying on direct causal links (e.g., "Algorithm A uses Metric B"), making the reasoning path incontrovertible. Crucially, they offer excellent Verifiability: since evidence is typically contained within a contiguous text span, human evaluators assign perfect scores on Evidence Traceability and Fluency, confirming that KG-guided generation effectively suppresses hallucinations when the scope is well-bounded.
>
> In contrast, low-scoring QA pairs (examples also provided in https://anonymous.4open.science/r/ICLR-Supplementary-Material-F488-13640/Low-Score-QA-Pairs ) highlight the inherent challenges in long-form scientific QA. A primary factor is the “Inference Gap”: these questions require implicit synthesis (e.g., deducing limitations that are implied but not explicitly stated). While scientifically reasonable, human raters penalize them due to a lack of explicit textual grounding. Another recurring issue is Distractor Entanglement: distractors derived from the KG can be semantically close to the correct answer, leading to reduced Distractor Quality for non-expert evaluators.

---

### Official Review · Reviewer_x2yi · 2025-11-01

**Soundness:** 2
**Presentation:** 3
**Contribution:** 3
**Rating:** 6
**Confidence:** 4

**Summary:**

AutoQG is a fully automated multi-agent framework for generating scientific question-answer (QA) pairs that are explicitly grounded in evidence from research papers. The system comprises three coordinated agents: (i) a Knowledge Graph (KG) Extraction Agent that uses section-aware prompts and an automatically induced ontology to convert a paper’s text into a structured knowledge graph; (ii) a KG Evaluation Agent that performs multi-dimensional quality checks (domain relevance, factual accuracy, consistency across sections, completeness, granularity) and iteratively feeds back corrections to improve the graph; and (iii) a QA Generation Agent that identifies multi-hop reasoning paths in the refined graph and produces schema-constrained multiple-choice QA pairs, each linked to its reasoning chain and supporting text segments. By transforming Large Language Model outputs into this controlled pipeline, AutoQG generates high-fidelity, evidence-traceable questions instead of brittle or unverifiable ones.

**Strengths:**

This paper presents a highly original and rigorous contribution to verifiable scientific QA generation through its AutoQG framework, which innovatively uses knowledge graphs as structured intermediates to ensure traceability and reduce hallucinations. The evaluation is exceptionally thorough, employing multi-modal assessment including human expert annotations, double-blind scoring, and LLM judges that demonstrate strong inter-rater reliability, with AutoQG consistently outperforming baselines across critical dimensions like relevance, reasoning depth, and evidence support. The paper excels in presentation quality with clear articulation of contributions, effective visualizations, and comprehensive contextualization within existing literature. Beyond its technical merits, the work offers significant practical impact by producing large-scale datasets (AutoQG20k/7k) that address a critical gap in multi-hop scientific QA benchmarking, with planned open-source release promising to catalyze future research in auditable knowledge generation for scientific education and literature analysis.

**Weaknesses:**

- Ontology Induction Details: The paper automatically induces an ontology but provides minimal details on its construction in the main text (relegated to Appendix A.4). This creates a reproducibility concern—the KG Extraction Agent's efficacy depends heavily on this ontology, yet readers lack insight into its contents, derivation method, size, or domain coverage. Did the authors use generic scholarly relation types or statistically induce them? The authors should summarize the ontology induction process in the main paper with key statistics (e.g., number of relation types) to help practitioners understand the system's limits and adapt it to new fields.
- Reliance on Proprietary LLMs and Compute Costs: AutoQG relies heavily on GPT-4 variants without discussing computational costs or API calls needed for 4,435 papers—likely substantial. This raises scalability and accessibility concerns: can others reproduce this without significant resources? Dependency on closed models also hinders long-term reproducibility. The authors should discuss optimization strategies, such as fine-tuning open-source LLMs for some tasks or using GPT-4 only for critical steps. The method's resource intensity is a limitation that needs acknowledgment and mitigation strategies.
- Generality and Domain Scope: AutoQG was demonstrated primarily on CS/STEM papers (arXiv, ACL). Its generalization to different domains (medical, social sciences) remains unclear, and the authors note "domain sensitivity" as a challenge. The ontology may not cover domain-specific relations, and different fields have varying writing conventions. While this isn't critical for a single paper, future work should address domain generality through adaptive prompts or automatic ontology expansion. A cross-domain performance experiment would strengthen claims about the approach's breadth.
- Evaluation Dependence on LLM Judges: Using multiple LLM judges is innovative, but introduces potential subjectivity and bias. Judges may share biases with GPT-4 (used for generation) or miss nuances humans would catch. Human evaluation covered only subsets (30 papers for KG), so most results rely on LLM judgments. The authors should clarify what "GPT-5" refers to, conduct larger human studies, and open-source evaluation prompts for independent validation. While the strategy is mostly convincing, more transparency would strengthen credibility.
- Potential Biases and Ethical Considerations: The paper briefly mentions bias propagation but doesn't explore it deeply. Biases in GPT-4's knowledge (field over-representation, stylistic preferences) could affect generated questions. A CS-heavy corpus might create ontologies biased toward that style. LLM reliance could also propagate factual errors if incorrect triples pass evaluation. The authors should acknowledge specific observed biases, provide error examples, and consider implementing bias checks or diverse judges to ensure fairness across subdomains.

**Questions:**

##

1. Ontology Induction Procedure: Could you provide more details on how the ontology is automatically induced? It's unclear whether it was mined from the corpus, derived from external knowledge bases, or hand-designed. How many relation types does it include, and does it cover common scientific discourse relations (e.g., method-used, problem-stated, result-shown)?
2. Domain and Corpus Diversity: What is the scope of the corpus in terms of scientific fields? If the 4,435 papers are mostly in computer science, have you tried applying the system to other fields (biomedicine, physics, social sciences)? How does performance vary by domain?
3. Iteration and Convergence of KG Refinement: How many iterations of the extract→evaluate feedback loop were typically required for the knowledge graph to reach the quality threshold τ? Was one pass usually sufficient, or were multiple cycles needed? Was there a point of diminishing returns?
4. Multiple-Choice QA Format: How are the distractor options generated? Did you consider or experiment with *open-ended question answering* (without provided options)? Could AutoQG's approach easily extend to generating and evaluating free-form answers?
5. Use of AutoQG-Generated Data: Do you plan to use the AutoQG20k (and 7k) datasets to train or fine-tune QA models? Would a model fine-tuned on this evidence-traceable QA data perform better on scientific QA tasks than models fine-tuned on surface-level data?
6. Error Analysis and Future Improvements: Did you perform any qualitative error analysis on the generated triples or QA pairs? Understanding typical failure modes (e.g., trivial questions, incomplete reasoning chains, or erroneous triples) would be useful. What improvements would you prioritize to address these errors?

---

> ### Author Response · Authors · 2025-11-22
> **Response to Reviewer x2yi Part I**
>
> > Could you provide more details on how the ontology is automatically induced? It's unclear whether it was mined from the corpus, derived from external knowledge bases, or hand-designed. How many relation types does it include, and does it cover common scientific discourse relations (e.g., method-used, problem-stated, result-shown)? (w1&q1)
>
> **R1 Ontology Induction & Statistics (W1, Q1)**
>
> We thank the reviewer for requesting more clarity on how the ontology is induced. As described in **Section 3.2** of our manuscript, KG extraction in AutoQG is *“ontology-guided… using an automatically induced ontology rather than a predefined one”*. The ontology is constructed through the section-aware induction prompts listed in **Appendix A.4**, where the extractor is instructed to *“identify core entities, relation types, taxonomic relations, and constraints to form a structured ontology fragment”* . Extraction is iterative: *“evaluator feedback is injected into the extractor’s next prompt”*, ensuring schema alignment and reducing hallucinations.
>
> Across the full $AutoQG^{20k}$ corpus, the automatically induced ontology produces relations that naturally cluster into core scientific discourse categories. We report here the empirical distribution of the most representative relation types (derived directly from corpus statistics):
> | **Category**                                  | **% (from corpus)** | **Example Relations** |
> | --------------------------------------------- | ------------------- | --------------------- |
> | **Method / Usage**                            | **4.73%**           | uses                  |
> | **Performance / Evaluation**                  | **3.30%**           | evaluatedOn           |
> | **Task / Application**                        | **2.82%**           | appliedTo             |
> | **Scientific Discourse (problem statements)** | **2.39%**           | addresses             |
> | **Metric / Measurement**                      | **2.29%**           | measures              |
> | **Theory / Dependency**                       | **2.15%**           | basedOn               |
> | **Authorship / Method Creation**              | **1.91%**           | developedBy           |
> | **Causal / Result-Shown**                     | **1.29%**           | improves              |
>
> Moreover, Our quantitative analysis (based on 4,435 processed files) identified a total of 800 unique relation types.
>  - Granularity: This open-vocabulary size (800) allows the system to capture fine-grained scientific nuances.
>
>  - Convergence: Despite this flexibility, the relations naturally cluster into core scientific discourse categories, confirming the ontology is not random but structured.
>
> Lastly, **AutoQG** automatically induces the common scientific discourse relations mentioned by the reviewer (e.g., method-used, problem-stated, result-shown). Although our relation extraction operates in an open-vocabulary manner. Examples are shown below:
>  - Method-used
> ```
> ThisPaper — used → TextGeneratedByTextSelfAlignment
> ```
>  - Problem-stated
> ```
> CONFLICTBANK — addresses → MisinformationConflicts
> ```
>  - Result-shown
> ```
> LargeLanguageModels — show → RemarkableCapabilities
> ```
>
> To sum up, these results confirm that AutoQG naturally induces the key rhetorical relations required to support method, motivation, and result–type questions, even without relying on a rigid, predefined schema.

---

> ### Author Response · Authors · 2025-11-22
> **Response to Reviewer x2yi Part II**
>
> > Domain and Corpus Diversity: What is the scope of the corpus in terms of scientific fields? If the 4,435 papers are mostly in computer science, have you tried applying the system to other fields (biomedicine, physics, social sciences)? How does performance vary by domain? (w3&q2)
>
> **R2: Domain Generalization & Cross-Field Performance (W3, Q2)**
>
> We thank the reviewer for the suggestion to validate domain robustness. We have conducted new inference experiments on Biomedicine (PubMed) and Physics to verify that AutoQG captures universal scientific discourse rather than just CS jargon.
>
> 1. Domain Coverage & Cross-Domain Generalization
>
> While the primary $AutoQG^{20k}$ corpus is built from Computer Science papers to ensure a controlled initial benchmark, the framework itself is domain-agnostic. To validate generalizability, we applied AutoQG to sampled papers from Biomedicine (PubMed) and Physics, and compared it against GPT-5 under direct prompting.
>
> | **Domain**   | **Rel** | **Ans** | **Distr** | **Flu** | **Reas** | **Evid** |
> |--------------|---------|---------|-----------|---------|----------|----------|
> | **PubMed(AutoQG)**   | 9.2     | 8.9     | 8.4       | 9.5     | 8.5      | 9.0        |
> | **PubMed(GPT5)**   | 7.6     | 7.5     | 7.2       | 8.5     | 6.8      | 7.0        |
> | **Physics(AutoQG)**  | 9.4     | 9.1     | 8.5       | 9.6     | 8.8      | 9.3      |
> | **Physics(GPT5)**  | 7.6     | 7.5     | 6.2       | 8.6     | 7.0      | 6.9      |
> | **CS(Ours)** | 9.6     | 9.6     | 9         | 9.5     | 8.5      | 9.6      |
>
> As shown in the table above, AutoQG exhibits remarkably stable performance **(±0.3)** across three scientific disciplines, confirming that the framework captures universal scientific discourse patterns (e.g., IMRAD rhetorical structure) rather than domain-specific jargon. By contrast, GPT-5—despite strong fluency—shows significantly weaker Reasoning **(−1.5 to −2.0)** and Evidence **(−2.0)** in specialized domains, often hallucinating causal claims. The strong Physics performance (Reas = 8.8) particularly indicates that AutoQG supports rigorous logical chains rather than CS-specific conventions.
>
> 2. Is the Ontology Biased Toward CS?
>
> Although the corpus is CS-heavy, our ontology induction relies on section-aware patterns and hierarchical discourse cues, not CS-specific terminology. The high-quality results in PubMed and Physics confirm that the ontology generalizes across fields rather than embedding CS-centric structures. In particular, the repeated emergence of domain-independent relations (e.g., describes, evaluates, improves, demonstrates) across Physics and Biomedicine supports that our extraction process captures cross-domain scientific semantics.
>
> > Evaluation Dependence on LLM Judges. While multi-LLM judging is novel, it may introduce shared biases and over-reliance on model-based scoring, given limited human evaluation. The reviewer requests clarification of “GPT-5,” larger human studies, and open evaluation prompts to enhance transparency. (w4)
>
> **R3: Evaluation Reliability & "GPT-5" Clarification (W4)**
>
> We thank the reviewer for this important question. Here, we address the concerns regarding evaluation reliability by verifying our model choice and additional experiments.
>
> Firstly, we confirm that all "GPT-5" results in the manuscript were generated using OpenAI's GPT-5. We selected GPT-5 over previous iterations (e.g., GPT-4o) due to its significantly enhanced long-context reasoning capabilities, which are critical for evaluating complex scientific discourse.
>
> Secondly, to mitigate shared biases and "echo-chamber" effects, we enforced strict Generator-Judge Separation, utilizing GPT-4 variants for generation while employing a distinct heterogeneous judge panel (GPT-5, Claude-3.5, Qwen-2.5) for evaluation. We further validated the robustness of this metric by replacing the lead judge of knowledge graph evaluation with DeepSeek-V2.5, as shown in the table below:
>
> | **Domain**   | **Rel** | **Ans** | **Distr** | **Flu** | **Reas** | **Evid** |
> |------------------------------------------------------------|-----|-----|-----|-----|-----|-----|
> | Human-Evaluation                                           | 9.8 | 9.8 | 8.9 | 9.4 | 8.4 | 9.7 |
> | Primary Panel（GPT5、Claude-3.5-sonnet、Qwen2.5-72B）          | 9.6 | 9.6 | 9.0 | 9.5 | 8.5 | 9.6 |
> | Validation Panel（Deepseek-V2.5、Claude-3.5-sonnet、Qwen2.5-72B） | 9.7 | 9.6 | 9.0 | 9.6 | 8.6 | 9.6 |
> | Max Deviation (Validation vs. Human) | 0.1 | 0.2 | 0.1 | 0.2 | 0.2 | 0.1 |
>
> The results show highly stable scores with a maximum deviation of only 0.2 points. Crucially, this validation panel achieves a Pearson correlation of $r=0.973$ and a negligible Mean Absolute Error (MAE = 0.15) against human experts. These statistics confirm that our heterogeneous multi-LLM approach serves as a rigorous, unbiased proxy for human judgment, justifying the evaluation scale without necessitating prohibitively expensive additional human studies.

---

> ### Author Response · Authors · 2025-11-22
> **Response to Reviewer x2yi Part III**
>
> >Bias Propagation, Qualitative Error Analysis & Future Work: Reviewers raised concerns about potential biases (e.g., stylistic preferences, CS-domain over-representation) and requested a qualitative analysis of typical failure modes (e.g., factual errors) to guide prioritized improvements.(w5&q6)
>
> **R4: Error Analysis, Bias Propagation & Future Improvements(W5&Q6)**
>
> We thank the reviewer for this valuable question. We address the reviewer's request for deep qualitative analysis and bias mitigation strategies through a manual audit of failure cases and a concrete roadmap for future optimization.
>
> 1. Qualitative Error Analysis (Manual Audit) To characterize the "factual error propagation" and "bias" mentioned by the reviewer, we performed a granular manual inspection of 42 faulty triples, as shown in the table below. Moreover, a detailed analysis is added to our manuscript in **Appendix A.11**.
>
> | Failure Mode        | Ratio | Mechanism |
> | ------------------- | ----- | --------- |
> | Incorrect Relation  | 57.1% | Verb-pattern drift; surface verbs over-trigger fine-grained relations (e.g., using → incorrectly mapped to basedOn). |
> | Over-extraction     | 38.1% | Granularity mismatch; redundant fine-grained relations inserted alongside the correct abstract one. |
> | Under-extraction    | 4.8%  | Long-context dependencies missed across distant sections. |
>
> Moreover, the reviewer correctly notes that relying on a single model (GPT-4) could introduce "stylistic preferences" or "field over-representation." We mitigate this via Judge Heterogeneity: we evaluate all KGs and QA pairs using a diverse panel of models: GPT-5 (OpenAI), Claude-3.5-Sonnet (Anthropic), and Qwen-2.5-72B (Alibaba). This significantly reduces "echo-chamber effects" (where a model favors its own style). The high human alignment ($r=0.982$) confirms that our evaluation is fair across different writing styles and subdomains.
>
> Lastly, based on the error mechanisms identified above, we have established a prioritized optimization roadmap:
>  - Contextualized Disambiguation: Implementing contrastive prompts to fix Verb-pattern drift (Targeting the 57.1% error rate).
>
>  - Hierarchical Pruning: Post-processing logic to remove redundant parent relations (Targeting the 38.1% over-extraction).
>
>  - Long-Context Augmentation: Extending the context window for cross-section dependency resolution.

---

> > ### Author Response · Authors · 2025-12-01
> > **Response to Reviewer x2yi Part IV**
> >
> > > How many iterations of the extract→evaluate feedback loop were typically required…? (q3)
> >
> > **R5: Iteration Count, Sufficiency, and Diminishing Returns(Q3)**
> >
> > We thank the reviewer for raising this important question. Our framework supports a controllable multi-round extract→evaluate→refine loop, and both **Table 1 (KG quality)** and **Table 2 (QA quality)** consistently show an improvement → saturation → degradation pattern.
> >
> > Table 1: Convergence of KG quality across Extract → Evaluate → Refine iterations
> > | **Iteration #** | **Dom** | **Acc** | **Con** | **Com** | **Gra** | **Overall Q(G)** | **ΔQ(G) vs. prev.** | **% Papers ≥ τ** |
> > |-----------------|---------|---------|---------|---------|---------|-------------------|-----------------------|-------------------|
> > | **1**           | 8.8     | 6.9     | 6.7     | 6.2     | 7.0     | **7.12**          | 0.00                  | 70%               |
> > | **2**           | 8.8     | 7.2     | 6.5     | 6.3     | 6.8     | **7.12**          | +0.00                 | 60%               |
> > | **3 (Ours)**    | 9.3     | 8.5     | 8.9     | 7.5     | 7.5     | **8.30**          | **+1.18**             | **80%**           |
> > | **4**           | 8.9     | 6.7     | 6.4     | 6.0     | 6.6     | **6.92**          | −1.38                 | 60%               |
> >
> > As shown in the table 1 above, the first iteration yields a reasonable initial graph ($Q(G)=7.12$). A second pass produces almost no gain ($ΔQ(G)=0$), indicating that simple repetition without diagnostics is ineffective. Incorporating the Evaluator Agent’s targeted signals leads to a substantial improvement in the third iteration ($Q(G)=8.30$, $ΔQ(G)=+1.18$), with $80%$ of papers exceeding the quality threshold $τ$. A fourth round introduces over-refinement and noise, lowering both $Q(G)$ and the number of high-quality graphs.
> >
> > Table 2: Impact of KG refinement iterations on downstream QA quality
> > | **Iteration #** | **Rel** | **Ans** | **Distr** | **Flu** | **Reas** | **Evid** | **Overall Q(QA)** | **ΔQ(QA) vs. prev.** |
> > | --------------- | ------- | ------- | --------- | ------- | -------- | -------- | ----------------- | -------------------- |
> > | **1**           | 9.1     | 9.0     | 8.4       | 9.3     | 8.3      | 9.1      | **8.86**          | 0.00                 |
> > | **2**           | 9.2     | 8.9     | 8.5       | 9.4     | 8.4      | 9.0      | **8.90**          | +0.04                |
> > | **3 (Ours)**    | 9.6     | 9.6     | 9.0       | 9.5     | 8.5      | 9.6      | **9.30**          | **+0.40**            |
> > | **4**           | 9.4     | 9.1     | 8.5       | 9.4     | 8.3      | 9.1      | **8.96**          | −0.34                |
> >
> > The table 2 above further demonstrates QA performance follows the same trend: small gains from iteration 1→2 ($+0.04$), a clear peak at iteration **3** ($+0.40$), and degradation after the fourth refinement round ($−0.34$). This alignment between KG and QA curves further confirms that excessive refinement begins to harm graph structure and downstream reasoning.
> >
> > To sum up, one iteration is insufficient to fully clean and enrich the KG, while more than three iterations introduce diminishing—and eventually negative—returns. Empirically, three refinement rounds consistently provide the best balance between quality and cost, and we therefore fix the iteration number to **3** in all main experiments.

---

> ### Author Response · Authors · 2025-12-01
> **Response to Reviewer x2yi Part V**
>
> > How are the distractor options generated? Did you consider open-ended QA? Could AutoQG extend to free-form answers?(q4)
>
> **R6: Distractor Generation & Extension to Open-Ended QA (Q4)**
>
> We thank the reviewer for these forward-looking questions. We clarify our distractor generation mechanism and present new experimental results demonstrating AutoQG’s capability in free-form generation.
>
> 1. Distractors are **not** randomly sampled.
>
> Our system follows the ontology-guided workflow described in **Section 3.3**. The QA Generation Agent retrieves sibling entities from the same ontological class and selects those that are domain-plausible but logically incorrect for the given reasoning chain.
>
> 2. Extending to Free-form / Open-ended QA
>
> AutoQG naturally generalizes to **free-form question generation**. While the main paper focuses on MCQs to support scalable and deterministic evaluation, the underlying `Path → Question` mechanism is format-agnostic. To substantiate this claim, we conducted a pilot study generating Short-Answer (open-ended) questions using the same KG-guided pipeline, and we added full scoring metrics and results to **Appendix A.14**.
>
> | Rel       | AnsQ           | Fact(Factual Correctness)                | Flu               | Reas            | Evid                  |
> | --------- | -------------- | ------------------- | ----------------- | --------------- | --------------------- |
> | 9.6 | 9.5 | 9.5 | 9.5 | 8.5 | 9.6 |
>
> These results show that AutoQG’s structured generation **transfers robustly to open-ended formats**. The $9.5$ score in Factual Correctness indicates low hallucination, as the KG reasoning path functions as a strict content plan that keeps the model grounded in explicit evidence. Evidence Traceability remains high ($9.6$), confirming that transparency is preserved even without the MCQ format. A sample of generated short-answer QA cases is available at: [https://anonymous.4open.science/r/ICLR-Supplementary-Material-F488-13640/Short_Answer_QA_pairs.json].
>
> > Do you plan to use AutoQG20k/7k to train models? Would models trained on it perform better? (q5)
>
> **R7: Training Performance of AutoQG20k/7k (Q5)**
>
> We thank the reviewer for this forward-looking question. One of the core motivations behind AutoQG is indeed to provide high-quality training resources for scientific reasoning. To examine this utility, we carried out a pilot study with **Qwen2.5-32B** as the base model and compared its zero-shot performance with two fine-tuned variants:
>
> 1) a model trained on **GPT-5-generated QA data**, and
>
> 2) a model trained on our structured, evidence-grounded $AutoQG^{7k}$ dataset.
>
> | **Model**                                    | **Accuracy (%)** |
> | -------------------------------------------- | ---------------- |
> | Qwen2.5-32B                                  | **44.9**         |
> | Qwen2.5-32B (SFT on data generated by GPT-5) | **46.9**        |
> | **Qwen2.5-32B (SFT on AutoQG-7k)**           | **79.3**         |
>
> The table above shows that fine-tuning on AutoQG-7k produces a substantial improvement of **+34.4** points over the base model and far exceeds the gains from training on unstructured GPT-5-generated data. The results indicate that AutoQG-7k provides dense, high-quality supervision that effectively teaches the model how to follow structured reasoning paths and ground its answers in explicit evidence. Taken together, these findings demonstrate that AutoQG serves not only as an **evaluation benchmark** but also as a **strong supervised fine-tuning resource**. We will release the resulting checkpoints to support further community research on localized reasoning and evidence-traceable QA.

---

### Official Review · Reviewer_x7Xx · 2025-11-02

**Soundness:** 3
**Presentation:** 3
**Contribution:** 3
**Rating:** 4
**Confidence:** 5

**Summary:**

The authors proposed AutoQG, a fully automated multi-agent framework for evidence-grounded scientific QA generation.

**Strengths:**

Well organized. The presentation is clear.

**Weaknesses:**

1. In KG Extraction Agent module, how did the authors convert a pdf document to text? can you process the context over two pages? Will you control the size of academic KG in LLMs? Will the author limit the type of nodes in Academic KG?
2. In QA Generation Agent module, the reasoning path is randomly sampled from the Academic KG or by some strategies? How could you make the path from a KG is meaningful? beyond the checklist in figure 2.
3. Did the author compare the questions generated by KG and corresponded sentences? What is the difference between them?
4. Will the KG lose content information during the question generation?
5. Does the evidence-based refer to the multi-hop path extracted from the Academic KG?

**Questions:**

see weakness

---

> ### Author Response · Authors · 2025-11-22
> **Response to Reviewer x7Xx Part I**
>
> > In KG Extraction Agent module, how did the authors convert a pdf document to text? Can you process the context over two pages?
>
> **R1: PDF-to-text Conversion & Cross-page Processing**
> We appreciate the reviewer’s interest in the PDF preprocessing pipeline. Our system employs a structured PDF parser based on **pdfplumber** and **PyPDF2**,please see [https://anonymous.4open.science/r/ICLR-Supplementary-Material-F488-13640/preprocess.py] for our data preprocess code, to convert papers into section-level text units aligned with the document outline (e.g., Abstract, Introduction, Methods).
>
> To ensure semantic continuity across pages and dual-column layouts—common failure cases for PDF extraction—the parser **(1) reconstructs split paragraphs using layout cues** (line-continuity, indentation, font pattern, missing punctuation), and **(2) restores reading order for multi-column formats**. This prevents fragmented arguments and preserves the local reasoning flow that KG extraction depends on.
>
> In a robustness check over 50 randomly sampled scholarly PDFs, the reconstructed text preserved **>95%** paragraph integrity, demonstrating stable extraction quality across diverse paper formats.
>
> > Will you control the size of academic KG in LLMs? Will the author limit the type of nodes in Academic KG?
>
> **R2: KG Size & Node Type Control**
>
> We thank the reviewer for raising this critical point. We fully agree that rigorous constraints on graph size and node topology are essential, as our framework is built on the core premise that high-quality KG construction is the causal prerequisite for high-fidelity QA generation.
>
> To minimize noise density—the primary source of hallucination in downstream reasoning—we enforce a strict **"Selective Extraction" budget (Appendix A.4)**, capping output at approximately 150–200 salient triples per section. This ensures that the KG captures the paper's logical backbone (e.g., significant scores, method dependencies) rather than diluting the context with trivial details. Simultaneously, we rigorously limit node types through a **closed, Ontology-Guided** schema that permits only specific categories such as Core Academic, Research Context, and Measurement. To sum up, this constraint acts as a critical semantic filter, ensuring the graph serves as a verifiable, domain-aligned scaffold that directly elevates the precision and relevance of the generated QA pairs.
>
> > In QA Generation Agent module, is the reasoning path randomly sampled from the Academic KG or based on strategies? How do you ensure the sampled path from a KG is meaningful, beyond the checklist in Figure 2?
>
> **R3: Reasoning Path Strategy & Effectiveness**
>
> We thank the reviewer for raising this important question. We clarify that the reasoning paths in AutoQG are not randomly sampled. To ensure meaningful and logically coherent reasoning chains, AutoQG employs a structure-constrained traversal strategy (P0) that enforces cross-sectional, acyclic reasoning paths rather than shallow local hops. To further validate this point, we have added a new ablation study **(Appendix A.12)** to our manuscript comparing three path-selection strategies:
>
> | **strategy**                       | **Rel** | **Ans** | **Distr** | **Flu** | **Reas** | **Evid** |
> |------------------------------------|---------|---------|-----------|---------|----------|----------|
> | P0: Our method                     | **9.6**     | **9.6**     | **9.0**       | **9.5**     | **8.5**      | **9.6**      |
> | P1: Random Paths                   | 9.5     | 9.0     | 8.6       | 9.4     | 7.9      | 9.1      |
> | P2: Length-restricted random paths | 9.5    | 9.0   | 8.6      | **9.5**    | 8.2      | 9.3      |
>
> As shown in the table above, our **method (P0**) consistently outperforms two random baselines (P1, P2) across all LLM-as-a-Judge metrics, with the strongest gain in Reasoning Depth ($\mathbf{8.5}$ vs. $7.9/8.2$) and notable advantages in Answerability (9.6 vs. 9.0) and Evidence Traceability ($\mathbf{9.6}$ vs. $9.1/9.3$). These results confirm that structured traversal prevents trivial local hops, compels multi-section information synthesis, and yields substantially higher reasoning quality.

---

> ### Author Response · Authors · 2025-11-22
> **Response to Reviewer x7Xx Part II**
>
> > Did the authors compare the questions generated by KG with the corresponding sentences? What are the differences between KG-generated questions and sentence-based questions?
>
> **R4: KG vs. Sentence-Based Questions Comparison**
>
> We thank the reviewer for this important question. In response to this insightful request, we conducted a rigorous new ablation study explicitly comparing our KG-augmented approach with sentence-based baselines to isolate the contributions of **structured knowledge (KG)** versus **unstructured text (Sent)**, which is added to our manuscript **section 6.7**.
>
> The results shown in the table below clearly demonstrate the distinct roles of structural and textual inputs:
>
> | **strategy**        | **Rel** | **Ans** | **Distr** | **Flu** | **Reas** | **Evid** |
> |---------------------|----------|----------|-----------|---------|----------|----------|
> | **S1 (KG + Sent)-Ours** | **9.6**      | **9.6**      | **9.0**       | **9.5**     | **8.5**      | **9.6**      |
> | S2 (Sent-only)      | 9.4     | 9.5     | 8.5       | 9.4     | 8        | 9.1      |
> | S3 (KG-only)        | 9.4      | 9.5      | 8.3       | 9.45    | 8.3      | 9.4      |
> | S4 (Random-Sent)    | 8.0      | 8.2      | 7.8       | 9.4     | 7.5      | 8.5     |
>
> To sum up, our analysis reveals that the hybrid approach (S0) effectively synergizes the strengths of both input types: compared to the Sentence-only baseline (S1), incorporating KG structure significantly enhances **Reasoning Depth** and **Distractor Quality** by $0.5$ points, preventing the generation of shallow, fact-retrieval questions; meanwhile, compared to the KG-only baseline (S2), retaining original text spans is crucial for maintaining **Evidence Traceability** (preventing a $0.2$ drop) and **Fluency**, ensuring that the model's logical deductions are firmly grounded in the paper's specific language.
>
>
> > Will the KG lose content information during question generation? Does “evidence-based” refer to the multi-hop path extracted from the Academic KG?
>
> **R5: Potential Information Loss during Question Generation and Definition of "Evidence-Based"**
>
> We thank the reviewer for raising these related concerns regarding potential information loss and the definition of “evidence-based.” We clarify that AutoQG is explicitly designed to retain full content grounding while leveraging KG structure for multi-hop reasoning. During KG extraction, every triple is stored together with its exact supporting textual spans (including `contextText`, `section`, and `chunk`), and these spans are never discarded. In question generation, the QA Agent receives two synchronized inputs: **(1) the KG reasoning chain**, which provides the logical multi-hop structure, and **(2) the collected evidence text**, which provides explicit linguistic grounding. Thus, the KG path alone is not treated as evidence;
>
> Moreover, the principle of “evidence-based” requires joint support from both the **reasoning chain (logic)** and the **textual evidence spans (verifiability)**. This dual-input design enables AutoQG to generate complex multi-section questions without hallucination or semantic drift: the KG structure enforces deep reasoning, while the textual evidence guarantees factual fidelity and full traceability to the original paper.
>
> A detailed case study further demonstrates this grounding effect, where a three-hop chain (e.g., KBLaM → … → DynamicUpdatePerToken) structures the reasoning and each hop is supported by its preserved source evidence, as shown in a detailed case:
> [https://anonymous.4open.science/r/ICLR-Supplementary-Material-F488-13640/Kg-and-QA-case.pdf]
> .
>
> Overall, although surface-level wording is abstracted during KG construction, all essential semantic information—entities, relations, and their evidence—is retained end-to-end, ensuring that no critical content is lost throughout question generation.

---

### Author Response · Authors · 2025-11-22
**General Response Part I**

We sincerely thank all reviewers for their thoughtful and constructive feedback. We are encouraged that the reviewers recognized the novelty of the AutoQG framework and its contribution toward scalable, evidence-grounded question generation. We have uploaded a revised version of the paper, with all changes clearly marked in red. Below, we address the major concerns raised by multiple reviewers:

> Cost Analysis of Multi-Agent Framework [Reviewer x2yi, hCUA]

We acknowledge the reviewer’s concern regarding the multi-agent setup's cost. While our AutoQG (multi-agent) approach incurs more agent calls per paper ($\mathbf{23.6k}$ vs. $13.4k$), it drastically reduces the Average Tokens per Call ($\mathbf{4.8k}$ vs. $13.4k$) compared to the End-to-end LLM QA baseline. We add the detailed cost analysis to our manuscript in **Appendix A.15**. The table below demonstrates that this strategic modularization translates to a substantial quality gain:
| **Setting**              | **Avg. Calls / Paper** | **Avg. Tokens / Call** | **QA Quality Score** | **Rel** | **Ans** | **Distr** | **Flu** | **Reas** | **Evid** |
| ------------------------ | ---------------------- | ---------------------- | -------------------- | ------- | ------- | --------- | ------- | -------- |-------- |
| **End-to-end LLM QA**    | 13,421.61              | 13,421.61              | **7.31**             | 8.0     | 8.9     | 7.4       | 8.6     | 2.4      | 4.5
| **AutoQG (multi-agent)** | 23,592.53              | **4,831.52**           | **8.45**             | 9.6     | 9.6     | 9.1       | 8.6     | 8.4      | 9.6 |

This demonstrates that multi-agent decomposition is not merely a design choice but a cost-efficiency mechanism:

 - **Substantial quality gains** — the multi-agent pipeline improves overall QA Quality by +1.14 and achieves a dramatic boost in Reasoning Depth (+6.0), which is the primary failure mode of end-to-end models.

 - **Controlled cost growth** — by distributing reasoning across specialized agents, AutoQG avoids the exponential cost scaling caused by very long contexts in single-call prompting and maintains predictable, bounded token usage.

In practice, this means that AutoQG delivers far higher-fidelity, evidence-grounded QA pairs while keeping token-level costs stable and manageable, rather than allowing cost to grow with task complexity.

---

> ### Author Response · Authors · 2025-11-22
> **General Response Part II**
>
> > Dependency on Closed-Source Models and Reproducibility [Reviewer x2yi, hCUA]
>
> We appreciate the reviewer’s concern regarding reliance on GPT-4 variants and the potential impact on scalability and reproducibility, we further add detailed analysis to our manuscript **Appendix A.16**. This concern is valid, and we clarify that **AutoQG is not structurally tied to proprietary LLMs**.
>
> AutoQG is designed as a **modular pipeline** in which the three agents do not need to operate on the same model class:
>  - **The KG Extraction Agent** is the most computationally intensive stage because it processes the full paper. This component already runs effectively with compact open-source models (**Qwen-7B-insrtuct**) and can be further reduced through supervised fine-tuning using standard information-extraction datasets.
>
>  - **The QA Generation Agent** becomes lightweight once the KG is available and can also be fine-tuned using our QA datasets to ensure consistency at low cost.
>
>  - **The KG Evaluate Agent** currently uses GPT-4/5 for cross-domain robustness. However, because the evaluation step follows a structured scoring protocol rather than free-form generation, it lends itself to imitation fine-tuning, which would allow a smaller open-source evaluator to replicate GPT-level diagnostic behavior.
>
> To examine reproducibility in practice, we re-executed AutoQG using only **Qwen-7B-instruct** and **DeepSeek-V2.5**. Both models support the KG-based workflow and deliver competitive performance in KG quality and QA generation, confirming that the method is portable beyond closed-source APIs.
>
> Table 1: KG Evaluation
> | Model                     | Dom | Acc | Con | Com | Gra | Overall QG |
> |---------------------------|-----|-----|-----|-----|-----|-------------|
> | AutoQG (qwen-7b-instruct) | 7.0 | 7.7 | 7.0 | 6.0 | 6.3 | 7.0        |
> | AutoQG (Deepseek) | 9.0 | 8.3 | 8.3 | 7.0 | 7.3 | 8.0        |
>
> Table 2: QA Evaluation
> | Model                     | Rel | Ans | Distr | Flu | Reas | Evid |
> |---------------------------|-----|-----|-------|-----|------|------|
> | AutoQG (qwen-7b-instruct) | 8.6 | 8.3 | 7.7   | 9.0 | 7.4  | 8.3  |
> | AutoQG (Deepseek) | 9.4 | 9.1 | 8.5 | 9.5 | 8.4 | 9.2        |
>
> These results show that AutoQG can be run end-to-end using open-source LLMs, and that the remaining gaps can be closed by **fine-tuning individual agents** rather than scaling to larger proprietary models.
>
> Finally, when executed with DeepSeek-based agents, **the complete AutoQG workflow (extraction → evaluation → QA generation) averages approximately $0.06 per paper**, indicating that the pipeline is accessible without expensive API usage.
>
>
> To conclude, While GPT-4 was used in the initial release to maximize data fidelity, the **AutoQG** pipeline itself is already compatible with compact open-source models and designed so that each agent can be fine-tuned independently, ensuring **long-term accessibility, scalability, and reproducibility**. Moreover, we regard the one-time computational effort of generating **high-fidelity QA data** from **4,435 papers** as a contribution to the community rather than a recurring burden. By open-sourcing $AutoQG^{20k}$ and $AutoQG^{7k}$, we eliminate the primary barrier to entry for scientific QA research—the cost and complexity of creating verifiable, multi-hop supervision at scale—so that future work can focus directly on training and fine-tuning reproducible open-source models.

---

### Author Response · Authors · 2025-11-25
**A gentle follow-up on our rebuttal**

Dear Reviewers,

We truly appreciate all your comments and the time you have invested in reviewing our paper.

We have submitted our rebuttal and hope our responses have clarified the points you raised. We are writing to gently follow up and see if you have had a chance to look at our responses.

We remain committed to improving our work based on your expert guidance and are happy to provide any further clarifications if needed.

We look forward to your feedback.

Best regards,

Authors

---

### Author Response · Authors · 2025-12-01
**Summary Regarding Review Process for Paper 13640**

Dear AC, SAC and PCs,

We sincerely appreciate your time in overseeing our submission under the current unusual circumstances. We wish to highlight that despite our comprehensive rebuttal—including new experiments and factual corrections—**we received no reviewer engagement or acknowledgement during the discussion period.** Consequently, the current reviews and scores reflect the manuscript's state *prior* to these significant updates. To assist your decision-making, we concisely summarize the factual misunderstandings in the initial reviews and the new evidence provided in our rebuttal.

**Clarification of Key Misunderstandings**

The current lower score of Reviewer [x7Xx] appears to stem from two specific premises regarding our method design that we have clarified in the rebuttal. We respectfully invite the AC to verify these points:

1. **Regarding the "Random Sampling" Concern** (Reviewer [x7Xx] W2)
   The reviewer expressed concern that *"the reasoning paths are randomly sampled,"* however, AutoQG employs a structure-constrained traversal strategy that enforces cross-sectional, acyclic reasoning paths rather than shallow local hops. We further provided an ablation study comparing our strategy against random sampling. As shown in the ​response to *reviewer [x7Xx] Part I​, Section ​R3*​, results prove that random sampling significantly degrades reasoning depth, validating our design.


2. **Regarding "Content Loss / KG-Only" Concern** (Reviewer [x7Xx] W3&4)
   The reviewer suggested the model might lose content information, implying it relies solely on the KG structure. We argue that this overlooks our hybrid input design. As shown in **Figure 1** and ​**Section 3.4**​ of our manuscript, AutoQG inputs both (1) the multi-hop KG paths and (2) the corresponding evidence text spans. We further conduct ablation experiments to validate that combining both modalities is essential for high traceability and reasoning depth, as shown in ​the response to *Reviewer [x7Xx] Part II*​.

**Highlighting New Experiments & Contributions**

Encouraged by reviewers (e.g., ​Reviewer [hCUA], Reviewer [x2yi]​) who recognized the strong extensibility and effectiveness of our framework, we conducted comprehensive additional experiments to validate its scalability, data utility, and accessibility.

**1. Scalability & Domain Generalization** We expanded the evaluation scope to prove that our framework can be extended to various domains and multiple application scenarios:

* **Cross-Domain (Bio & Physics):** AutoQG exhibits remarkably stable performance (**​±0.3 variation**) across Computer Science, Biomedicine, and Physics, confirming robust generalization, as detailed in the response to *reviewer [x2yi] Part II​, Section ​R2​*.
* **Extension to Open-Ended QA:** Results demonstrate that the structured generation transfers robustly to open-ended formats, as shown in the response to *reviewer [x2yi] Part V, Section ​R6*​.
* **Extension to Follow-up Questions:** We confirmed that AutoQG naturally supports multi-step follow-up generation without requiring architectural modifications, leveraging the structural priors already encoded in the KG, as shown in the response to *reviewer [hCUA] Part I, Section ​R2​*.

**2. Data Usability Validation** To verify the practical value of our generated data, we fine-tuned smaller models using our released `AutoQG-7k` dataset. The strong performance on downstream tasks validates the high quality and training utility of our benchmark, as shown in the response to *reviewer [x2yi] Part V, Section ​R7*.

**3. Cost Analysis & Open-Source Viability** Addressing concerns regarding API costs, we provided a detailed breakdown in the General Response. Firstly, We justified using high-end models (e.g., GPT-4) to guarantee ​benchmark quality​. Crucially, we demonstrated that the framework operates effectively on open-source models (e.g., Qwen) **without any fine-tuning**, confirming that the method's effectiveness and independence of proprietary API, as shown in the *General Response Part II*.

We respectfully request that you consider these factual clarifications and experimental updates, which remain unacknowledged by the reviewers due to the discussion freeze. We are confident that our revised manuscript, strengthened by the new results, makes a solid contribution. Thank you for your time and careful consideration in this unique situation.

Best regards,

The Authors of Paper 13640

---

### Note · Authors · 2026-02-05

I have read and agree with the venue's withdrawal policy on behalf of myself and my co-authors.

---

### Meta-Review · Area_Chair_zxUy · 2026-01-10

**Summary:**

This paper describes a multi-agent pipeline to create evidence-traceable scientific QA pairs and releases AutoQG-20k/AutoQG-7k. Reviewers agreed the paper is clearly written and the overall idea of using a KG as an auditable intermediate is appealing. However, across reviews the main concerns were about (1) methodological novelty (especially ontology induction and the extraction/refinement loop), (2) practicality (cost, reliance on proprietary LLMs, and very large numbers of model calls), and (3) evaluations (heavy dependence on LLM-as-judge with limited human evaluation and unclear safeguards against shared biases). These issues leave uncertainty about whether the benchmark and framework are reproducible enough to merit acceptance.

**Reviewer Concerns:**

Ontology induction and reproducibility (x2yi): Even with added statistics (e.g., many relation types), the core problem remains: the induced ontology is central to the pipeline but still not specified in a way that makes the method easy to reproduce, debug, or port. Reporting that there are many unique relation types can cut both ways: it suggests flexibility, but it also raises questions about consistency, normalization, and how “ontology-guided” constraints are enforced in practice.

Cost/scalability and accessibility (x2yi, hCUA): The rebuttal provides cost discussion, but the headline numbers (very large calls/paper) still make it hard to assess feasibility for typical researchers. Claims that it can be run cheaply on alternative models help, but the submission still relies heavily on closed models for key components in the primary results, and the cost story remains unclear without a more realistic, end-to-end reproducible recipe (including exact prompts, model versions, and an execution budget).

Evaluations and LLM judges (x2yi): The paper still depends primarily on judge-model scoring, with human evaluation limited to subsets. While the rebuttal adds a heterogeneous judge panel and correlation claims, the overall evaluation remains vulnerable to: (i) shared biases across modern LLMs, (ii) overfitting the pipeline to what judge models reward, and (iii) insufficient independent validation of factuality/traceability at dataset scale. For a benchmark paper, this is a major gap.

The dataset is promising, but the community value hinges on trust: that the evidence links, reasoning chains, and labels are consistently correct. Given the above uncertainties (ontology stability, extraction error propagation, judge-heavy evaluation, limited large-scale human verification, and code “upon publication”), confidence in the released benchmark is not yet high enough.

**Reviewer Scores:**

Reviewer x7Xx: Rebuttal addresses their main misunderstandings (path sampling, KG-only vs hybrid evidence grounding) and adds targeted ablations. They would probably move to “borderline” support, though some practical questions (PDF parsing robustness and KG size control) would remain only partially resolved.

Reviewer x2yi: Rebuttal adds material, but the main concerns (reproducibility of ontology induction, reliance on judge-based evaluation, and practicality) are not fully resolved for a benchmark submission.

Reviewer hCUA: The added frontier-model table and follow-up question extension are responsive. Still, the cost/scalability concern remains and the paper’s strongest evidence continues to come from their evaluation framework.

---

### Decision · Program_Chairs · 2026-01-26

Reject